# Polarimetric L-Band ALOS2-PALSAR2 for Discontinuous Permafrost Mapping in Peatland Regions

Ridha Touzi [1,*] , Steven M. Pawley [2] , Paul Wilson [1], Xianfeng Jiao [1], Mehdi Hosseini [1]
and Masanobu Shimada [3]

1 Canada Centre for Remote Sensing, 580 Booth Street, Ottawa, ON K1A0E4, Canada
2 Alberta Geological Survey, 4999-98 Avenue, Edmonton, AB T6B 2X3, Canada
3 Tokyo Denki University, Ishizaka, Hatoyama, Hiki, Saitama 350-0394, Japan
* Correspondence: ridha.touzi@canada.ca

**Abstract:** Recently, it has been shown that the long penetrating polarimetric L-band ALOS is very promising for boreal and subarctic peatland mapping and monitoring. The unique information provided by the Touzi decomposition, and the dominant-scattering-type phase in particular, on peatland subsurface water flow permits an enhanced discrimination of bogs from fens, two peatland classes that can hardly be discriminated using conventional optical remote sensing sensors and C-band polarimetric SAR. In this study, the dominant and medium-scattering phases generated by the Touzi decomposition are investigated for discontinuous permafrost mapping in peatland regions. Polarimetric ALOS2, LiDAR, and field data were collected in the middle of August 2014, at the maximum permafrost thaw conditions, over discontinuous permafrost distributed within wooded palsa bogs and peat plateaus near the Namur Lake (Northern Alberta). The ALOS2 image, which was miscellaneously calibrated with antenna cross talk ($-33$ dB) much higher than the actual ones, was recalibrated. This led to a reduction of the residual calibration error (down to $-43$ dB) and permitted a significant improvement of the dominant and medium-scattering-type phase ($20°$ to $-30°$) over peatlands underlain by discontinuous permafrost. The Touzi decomposition, Cloude–Pottier $\alpha$-H incoherent target scattering decomposition, and the HH-VV phase difference were investigated, in addition to the conventional multipolarization (HH, HV, and VV) channels, for discontinuous permafrost mapping using the recalibrated ALOS2 image. A LiDAR-based permafrost classification developed by the Alberta Geological Survey (AGS) was used in conjunction with the field data collected during the ALOS2 image acquisition for the validation of the results. It is shown that the dominant- and scattering-type phases are the only polarimetric parameters which can detect peatland subsurface discontinuous permafrost. The medium-scattering-type phase, $\phi_{s2}$, performs better than the dominant-scattering-type phase, $\phi_{s1}$, and permits a better detection of subsurface discontinuous permafrost in peatland regions. $\phi_{s2}$ also allows for a better discrimination of areas underlain by permafrost from the nonpermafrost areas. The medium Huynen maximum polarization return ($m_2$) and the minimum degree of polarization (DoP), pmin, can be used to remove the scattering-type phase ambiguities that might occur in areas with deep permafrost (more than 50 cm in depth). The excellent performance of polarimetric PALSAR2 in term of NESZ ($-37$ dB) permits the demonstration of the very promising L-band long-penetration SAR capabilities for enhanced detection and mapping of relatively deep (up to 50 cm) discontinuous permafrost in peatland regions.

**Keywords:** polarimetry; synthetic aperture radar; ALOS2; discontinuous permafrost; peatlands



## 1. Introduction

Recent climate warming has been pronounced in the Arctic and sub-Arctic regions, compared to global averages. Observed increases in surface air temperature have led to permafrost thaw, and accelerated warming may lead to decreases in near-surface terrestrial permafrost cover [1]. Climate warming is causing the initiation and expansion of abrupt

permafrost thaw (called thermokarst), which, even though it occurs at point locations, often causes much deeper permafrost thaw to occur more rapidly. Northern Alberta contains a significant component of discontinuous permafrost, which is distributed within wooded palsa bogs and peat plateaus that form part of a heterogeneous mosaic of nonpermafrost wooded bogs, fens, swamps, and other upland forest types. The permafrost distribution and ongoing degradation affect the peatland structure, hydrology, and vegetation [2] and has been linked with environmental changes including an increased stream runoff [3], greenhouse gas fluxes [1], and forest fire severity [4]. In addition, permafrost represents an important consideration for route planning and reclamation design because linear disturbances from seismic lines, pipelines, and winter roads result in the rapid and irreversible thawing of the underlying frozen peat [4].

However, to date, in Alberta, there has been a limited mapping or monitoring of permafrost at a scale sufficient for these purposes, with previous airphoto-based interpretations providing only a small-scale delineation of the forest-covered permafrost terrain. Recent work by the Alberta Geological Survey (AGS) has employed optical (Landsat-8) remote sensing and a GIS modelling approach for mapping discontinuous permafrost principally from LiDAR [5,6]. Early results have demonstrated that a relatively high classification accuracy can be achieved, and that permafrost is more extensive than previously identified even at relatively low latitudes (56.5°).

Cost-effective permafrost characterization and monitoring should be possible due to advances in the technology of earth observation satellites, and in particular L-band satellite SAR [7–13]. In particular, the long-penetration capabilities of L-band ALOS2-PALSAR2 should permit a large-scale mapping of discontinuous permafrost in peatland areas. Recently, we investigated the potential of the Touzi decomposition [14–16] and Cloude–Pottier incoherent target scattering decomposition (ICTD) [17,18], as well as the Freeman model-based decomposition (MBD) [19] for peatland classification using L-band polarimetric ALOS [20–22]. We showed that among all the parameters generated by the Touzi, Cloude–Pottier, and Freeman ICTDs, only the dominant-scattering-type phase generated by the Touzi decomposition [16,23] was sensitive to peatland subsurface water flow. This permitted an accurate discrimination of bogs from fens, two important wetland classes of similar vegetation that can hardly be discriminated by visible near-infrared (VNIR) satellites [24,25]. The combination of VNIR optical and all-weather polarimetric C-band SAR and (single or dual-pol) L-band SAR satellite information does not permit an accurate classification of peatlands either [21,22,24,25]. The detection of peatland subsurface water flow using L-band polarimetric ALOS allows for a clear discrimination of (open) bogs and (poor) fens using their different hydrological properties. This was demonstrated for boreal peatlands (in La Baie des Mines, and the Athabasca oil-sand exploration region) and subarctic peatlands located in the Hudson Bay at the Wapusk National Park [20–22,26]. The dominant-scattering-type phase ($\phi_{s1}$) provided by the Touzi decomposition, which permits the optimum exploitation of the long-penetrating L-band ALOS sensitivity to peatland subsurface water flow, should be an excellent candidate for the optimization of the polarimetric ALOS2 information in support of the enhanced detection of subsurface discontinuous permafrost in peatland regions. The medium-scattering-type phase ($\phi_{s2}$), which is not equal to the opposite of $\phi_{s1}$ for asymmetric scattering, can also provide complementary information for the enhanced characterization of peatland subsurface discontinuous permafrost. The potential of the dominant- and medium-scattering-type phases for subsurface discontinuous permafrost detection is confirmed in this study using an ALOS2 image collected over the Namur Lake region of Northern Alberta [27].

It is worth noting that the very promising results with the scattering-type-phase were obtained thanks to the excellent polarimetric calibration of ALOS-PALSAR [28,29] and the excellent PALSAR noise NESZ (noise-equivalent sigma zero of about −34 dB [29]), as discussed in [20–22]. Therefore, it is important to validate the data quality of the ALOS2-PALSAR2 images used in this study for subsurface discontinuous permafrost detection. The data quality investigation conducted in the following shows the importance of the

PALSAR2 recalibration in support of the enhanced detection of subsurface discontinuous permafrost in peatland regions.

In the following, the Touzi decomposition is briefly described in Section 2. The permafrost study site, the AGS permafrost classification [5] and the wetland classifications available at the site, as well as the field data collected during the ALOS2 acquisitions are described in Section 3. The data quality of the ALOS2 image collected at the study site is assessed in Section 4. It is shown that the recalibration of the ALOS2 image leads to a significant enhancement of the dominant- and medium-scattering-type phases in areas of low VH. The results obtained with the Touzi decomposition applied to the recalibrated ALOS2 image are presented in Section 5. A comparison of the dominant- and medium-scattering-type phases (($\phi_{s1}$) and ($\phi_{s2}$)) with the phase difference of HH and VV ($\phi_{HH}$-$\phi_{VV}$), the conventional multipolarization (HH, HV, and VV) channels, and the Cloude–Pottier ICTD's main parameters ($\alpha$ and H) is also conducted. It is shown that the medium scattering phase $\phi_{s2}$, combined with the extrema of the degree of polarization (DoP) and the Huynen maximum polarization return of the medium scattering, leads to the best identification of discontinuous permafrost areas. Finally, the results obtained with ALOS2 are discussed, and the requirement on NESZ is discussed for an optimum exploitation of long-penetrating L-band polarimetric SAR information in support of an enhanced mapping and monitoring of discontinuous permafrost and peatlands.

## 2. The Touzi Decomposition for a Unique Basis-Invariant Characterization of Polarimetric Target Scattering

The Touzi decomposition [14–16] was introduced for the optimum decomposition of coherent and partially coherent target scattering in terms of roll-invariant and unique target parameters. In contrast to the Cloude–Pottier ICTD [17,18], the decomposition uses a complex entity for an unambiguous target-scattering-type description. The scattering type's magnitude $\alpha_s$ and phase $\phi_s$ permit a unique and basis-invariant description of the target scattering type. The Huynen helicity is used for the assessment of the symmetric nature of the target scattering. Like the Cloude–Pottier ICTD, the Touzi decomposition is based on the characteristic decomposition of the coherency matrix $[T]$. The latter permits the representation of $[T]$ as the incoherent sum of up to three coherence matrices $[T]_i$ representing three different single scatterers, each weighted by its appropriate positive real eigenvalue $\eta_i$:

$$[T] = \sum_{i=1,3} \eta_i [T]_i \tag{1}$$

A roll-invariant coherent-scattering model, the Touzi scattering vector model (TSVM) [14], is used for the parametrization of the coherency eigenvectors in terms of unique target characteristics. Each single-target scattering $\vec{k}_i$ is represented as [14,16]:

$$\vec{k} = m \cdot ([R(\psi)] \otimes [R(\psi)]) \cdot \begin{bmatrix} \cos \alpha_s \cos 2\tau \\ \sin \alpha_s e^{j\Phi_s} \\ -j \cos \alpha_s \sin 2\tau \end{bmatrix} \tag{2}$$

where $[R(\psi)]$ is the rotation transformation matrix by the angle $\psi$, and $\alpha_s$ and $\Phi_s$ are the polar coordinates of the scattering type. $\tau$ and m are Huynen's maximum polarization helicity and return, respectively [30]. The target-scattering decomposition is conducted through an in-depth analysis of each of the three single-scattering eigenvectors ($i = 1,3$). Each scattering $i$ is represented in terms of five independent parameters: ($\eta_i$, $m_i$, $\alpha_{si}$, $\phi_{si}$, and $\tau_i$), where $\eta_i = span.\lambda_i$, and $\lambda_i$ is the normalized eigenvalue that measures the relative energy carried by the single scattering $i$.

It is worth noting that the representation of the scattering type in terms of the "symmetric" scattering-type polar coordinates ($\alpha_s$ and $\Phi_s$) leads to a unique presentation of the target scattering type, which is independent of the basis of the polarization, as recently shown in [31]. Consequently, $\alpha_s$ and $\Phi_s$ preserve the same value even when the decomposition is applied into a different basis of polarization (circular polarization, for example) [31].

This is not the case for the Cloude $\alpha$, which leads to a different scattering-type description whether the ICTD is applied with a circular polarization (leading to the Corr and Paladini ICTDs [32,33]), or with the Pauli basis polarization (i.e., Cloude–Pottier ICTD [17]), as discussed in [31].

The Touzi decomposition is assessed for discontinuous permafrost mapping in Section 5. The unique potential of the dominant- and medium-scattering-type phases ($\phi_{s1}$ and $\phi_{s2}$) for subsurface permafrost detection using PALSAR2 is also demonstrated.

## 3. Description of the Study Site, Permafrost, and Wetland Classifications Available, and ALOS2 Image Investigated

### 3.1. Study Site, Permafrost, and Wetland Classifications Available on the Site

The method described above was implemented in this study to determine the utility of polarimetric phase information extracted from an ALOS-2 (FP6-4) image for the characterization of discontinuous permafrost. The area of interest for this study was in the vicinity of Namur Lake, in Northern Alberta. The study area is located approximately 70 km West of Fort MacKay, Alberta. The Athabasca Oil Sands region, which includes Fort MacKay and Fort McMurray, Alberta, is a region known for extensive anthropogenic activities related to resource extraction, as well as extensive boreal peatlands. The area is underlain by discontinuous permafrost and lies at a latitude of approximately 57.25° north. The region's significant component of discontinuous permafrost is distributed within peat plateaus and wooded palsa bogs. These form part of a heterogeneous mosaic of nonpermafrost wooded bogs, fens, swamps, and other upland forest types.

Three classifications are available for our study site. First, a classification developed by the Alberta Geological Survey (AGS) who employed a remote sensing and GIS modelling approach for mapping discontinuous permafrost [5]. Under the assumption that discontinuous permafrost is located in bog plateaus with extensive caribou lichen, LiDAR and Landsat-8 data were combined for mapping peatlands and permafrost terrain [5]. Figure 1 presents the AGS classification of the study site. The classes found in the AWS classification include bog, collapse scar bog, fen, marsh, permafrost and water. The image scale, the four cardinal directions (north (N), south (S), east (E), west (W)) directions, and the coordinates that mark the geolocation are also indicated in Figure 1.

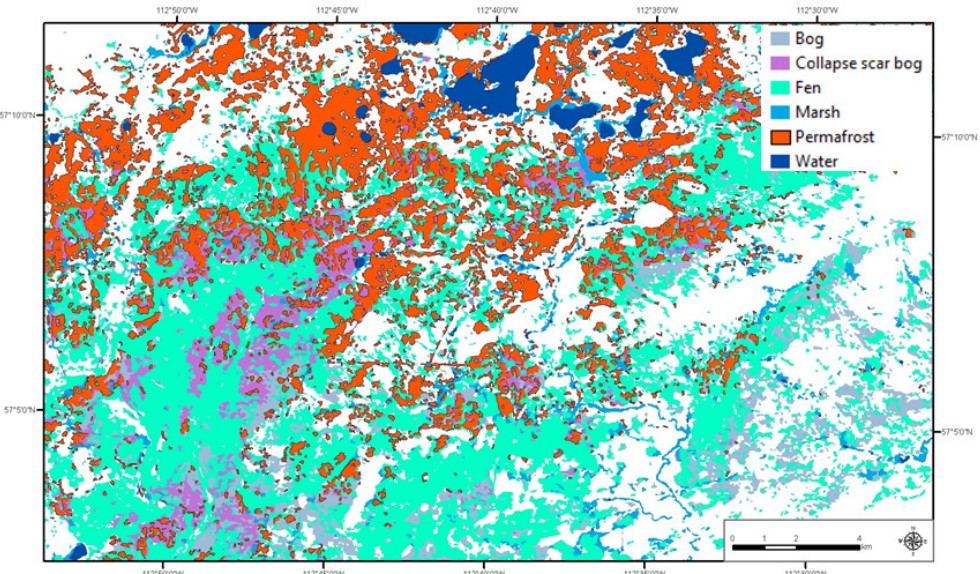

**Figure 1.** AGS Classification

Since the AGS classification was developed under the assumption that discontinuous permafrost is underlying peat plateaus covered with lichen, it is important to use, in addition to the AGS permafrost classification, a wetland classification that identifies

peatlands (treed bogs, bogs, and poor fens) and separates them accurately from upland forests and the conventional wetland classes. Two classifications were available for our study site located near Namur Lake: (1) the Alberta Ground Cover Classification (AGCC), a land-cover/land-use classification developed in 1990 by Alberta Environment and Sustainable Resource Development (ESRD, now called Agriculture and Forestry); (2) the Alberta Wetland Inventory (AWI) classification developed by the University of Alberta [34]. The AGCC classification was generated using Landsat5-TM images collected in 1990, a digital elevation model, and has been updated frequently using field data sampling. Figure 2 presents the AGCC of the Namur Lake study site. The image scale, the four cardinal directions (N, S, E, W), and the coordinates that mark the geolocation are also indicated in Figure 2. The main classes encountered at the study site are black spruce bog (sphagnum understorey) (6–100% tree cover), graminoid wetlands (sedges/grasses/forbs, less than 6% tree cover, less than 25% shrub), shrubby wetlands (willow and birch, less than 6% tree cover, more than 25 % shrub), and water (lake, pond, reservoir, river, and stream). Uplands are mainly dominated by closed white spruce forests.

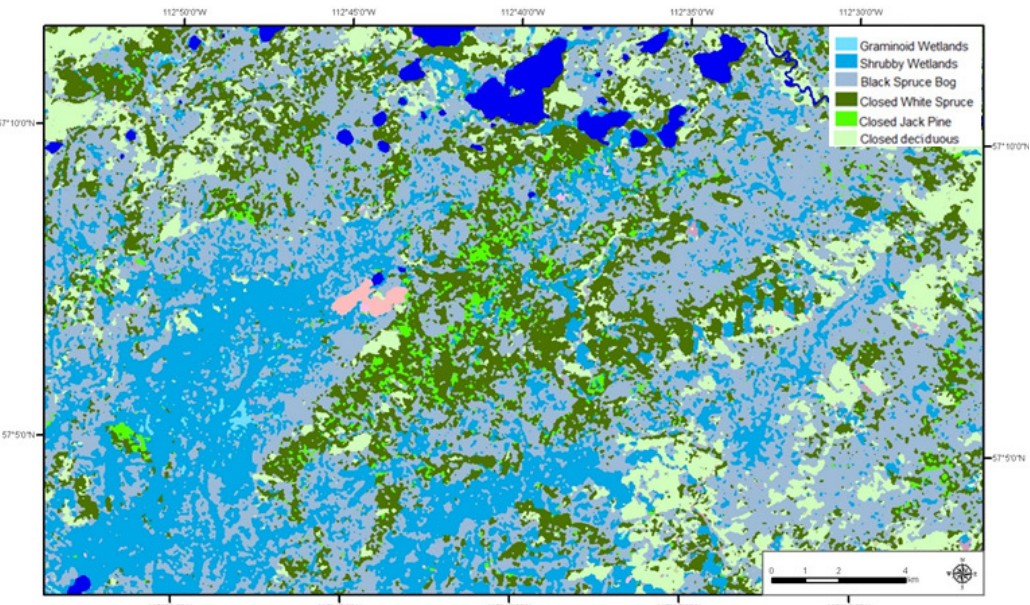

**Figure 2.** AGCC classification; burned areas are presented in light pink.

The Alberta Wetland Inventory (AWI) classification [34] was developed using aerial photographs. The AWI is based on the gross characteristic of vegetation visible from aerial photographs and has been validated using intensive field sampling. Figure 3 presents the AWI classification available at the Namur Lake site. The image scale, the four cardinal directions (N, S, E, W), and the coordinates that mark the geolocation are also indicated in Figure 3. The classification regroups all upland forests into one class named forest (white in Figure 3). The bog class represents treed bogs with 6 to 70% of tree cover. The fen class regroups open and treed fens with a low tree density (6%). Two additional classes are assigned to marsh and swamps.

It is worth noting that since the AWI is based on high-resolution aerial photographs, more details can be seen in the AWI than in the 30 m Landsat-based AGCC classification. As a result, the AWI can lead to a more accurate treed-bog class identification in comparison with the AGCC, as shown in [21] and confirmed in Section 5.3.

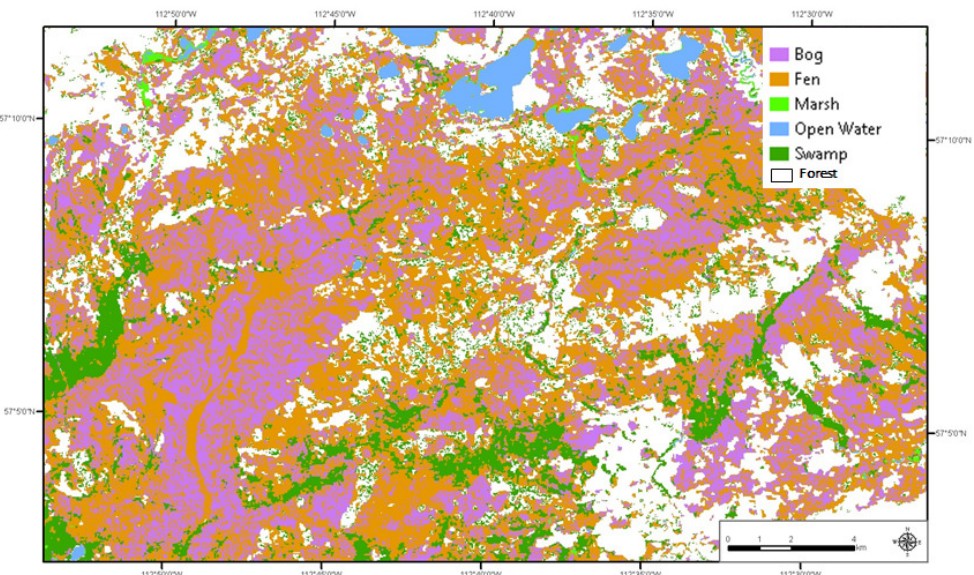

**Figure 3.** AWI classification.

### 3.2. PALSAR2 Image Investigated

In conjunction with JAXA, a polarimetric ALOS2 acquisition was planned on the 23 August at the maximum permafrost thaw conditions. Field data were collected during the week of the ALOS2 acquisition over discontinuous permafrost distributed amongst peat plateaus and wooded palsa bogs near Namur Lake. The polarimetric PALSAR2 image collected on 23 August 2014 with the highly sensitive PLR mode of about 4.4 m × 5.1 m resolution at FP6-4 (about 27 degrees of incidence angle) was used in this study. Figure 4 presents the multipolarization PALSAR2 image over the study site. The image scale, the four cardinal directions (N, S, E, W), and the coordinates that mark the geolocation are also indicated in Figure 4. Archived climate data from a nearby Environment Canada weather station confirmed that there was little precipitation in the days preceding the PALSAR2 acquisition (no rain for 10 days before the PALSAR2 image collection), an important determination considering that the backscattering from SAR signals had a strong correlation with water content and soil texture/composition due to its sensitivity to the dielectric properties of subsurface soils and surface water.

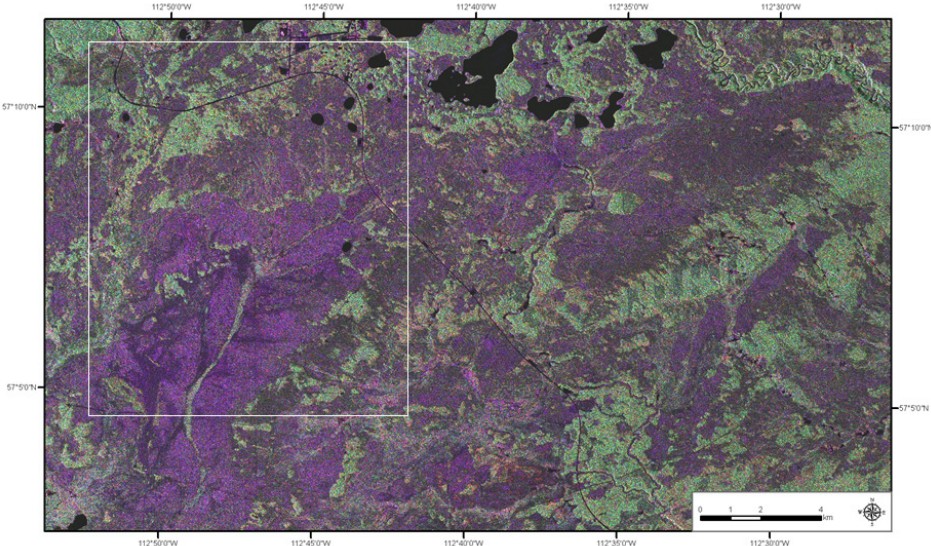

**Figure 4.** ALOS2 multipolarization image: HH (red), HV (green), and VV (Blue).

## 4. Recalibration of PALSAR2 Image for Optimum Detection of Subsurface Discontinuous Permafrost

### 4.1. Polarimetric PALSAR2 Image Calibration

After the correction of PALSAR2 transmit–receive antenna gain variations with the incidence angle [35,36], the following model can be used to express the voltage measurements as a function of the illuminated target-scattering matrix $[S]$ and the Faraday rotation angle $\Omega$ [10,37–39]:

$$[V] = \begin{bmatrix} 1 & \delta_1 \\ \delta_2 & F_1 \end{bmatrix}^T [S]_\Omega \begin{bmatrix} 1 & \delta_3 \\ \delta_4 & F_2 \end{bmatrix} \tag{3}$$

where the measured voltage matrix $[V]$ is given by:

$$[V] = \begin{bmatrix} V_{HH} & V_{HV} \\ V_{VH} & V_{VV} \end{bmatrix} \tag{4}$$

$[S]_\Omega$ is given as a function of the Faraday rotation angle $\Omega$ and the scattering matrix $[S]$ by:

$$[S]_\Omega = \begin{bmatrix} \cos\Omega & \sin\Omega \\ -\sin\Omega & \cos\Omega \end{bmatrix} [S] \begin{bmatrix} \cos\Omega & \sin\Omega \\ -\sin\Omega & \cos\Omega \end{bmatrix} \tag{5}$$

and the actual target scattering matrix $[S]$ is given by:

$$[S] = \begin{bmatrix} S_{HH} & S_{HV} \\ S_{VH} & S_{VV} \end{bmatrix} \tag{6}$$

In Equation (3), $[.]^T$ denotes the matrix transpose; $F_1$ and $F_2$ are the channel imbalances between the H and V channels to receive and transmit, respectively. $\delta_3$ and $\delta_1$ are the cross talks when a vertically polarized wave is transmitted and received, respectively. $\delta_4$ and $\delta_2$ are the cross talks when a horizontally polarized wave is transmitted and received, respectively.

The scattering matrix $[S]$ can be derived from Equation (3) using an estimation of the transmit–receive distortion matrices and channel imbalances. A first analysis of the data quality of the polarimetric ALOS2 image revealed a miscellaneous problem with the FP6-4 mode calibration. The distortion matrix applied included significant cross talks given in the following Table 1:

**Table 1.** PALSAR2 Transmit-Receive Antenna Cross-Talk Intensities (in dB).

| Cross-Talk | $\delta_1^T$ | $\delta_2^T$ | $\delta_3^R$ | $\delta_4^R$ |
|---|---|---|---|---|
| FP6-4 | $-32.27$ | $-33.80$ | $-36.59$ | $-35.47$ |

In fact, PALSAR2 is equipped with a highly isolated antenna with cross talks lower than $-40$ dB, as shown in [10,39–42] using different calibration approaches. The application of the distortion matrix correction with the cross talks given in Table 1 should lead to a significant residual error induced by the adopted transmit-antenna cross talks, $\delta_1 = -32.27$ and $\delta_2 = -33.80$ in Table 1.

It is worth noting that the FP6-4 calibration with the miscellaneous significant antenna cross talk ($-33$ dB) comfortably meets the CEOS cal-val requirements (cross talk lower than $-30$ dB) [43,44]. However, the scattering-type phase sensitivity to peatland subsurface permafrost might require a much lower residual calibration error, as demonstrated in this study. PALSAR2 recalibration using the actual antenna distortion matrix should lead to a significant reduction of the residual calibration error, and this permits a full exploitation of the excellent PALSAR2 NESZ (better than $-37$ dB [10,35,45]) for the enhanced detection of subsurface discontinuous permafrost, as discussed in the following.

### 4.2. Recalibration of PALSAR2 Image

L-band ALOS2-PALSAR2 polarimetric data are provided in terms of the Faraday rotation contaminated scattering matrix $[S_\Omega]$'s quad-pol voltage measurements. The PALSAR2 image recalibration included two steps:

1. Insert the transmit and receive distortion matrices (provided with the PALSAR2 data) in Equation (3) to derive the original voltage measurements.
2. Apply Equation (3) with the actual transmit–receive distortion matrices (cross talk lower than −40 dB [10,39–42]) to generate the recalibrated scattering matrix $[S_\Omega]$'s quad-pol voltage measurements.

The recalibration of FP6-4 with the actual PALSAR2 distortion matrices obtained in [40–42] led to a very weak residual error, less than −43 dB, as discussed in [39]. This should permit a full exploitation of the excellent PALSAR2 NESZ (−37 dB) and the long-penetration L-band polarimetric ALOS2 capabilities for the detection of the discontinuous permafrost underlying peatlands. Figure 4 presents the multipolarization PALSAR2 image. The AGS permafrost classification is presented in Figure 1. The AGCC and AWI classification are presented in Figures 2 and 3. The study area presented in Figure 4 is dominated by treed bogs according to the AWI classification of Figure 3 (bogs outlined in pink) and the AGCC classification of Figure 3 (black spruce bog in grey). The AWI and AGCC classifications were used in a complementary manner to the AGS classification to demonstrate the importance of the PALSAR2 image recalibration for an accurate mapping of peatlands and discontinuous permafrost. The impact of the ALOS2-PALSAR2 recalibration on the scattering matrix measurements and its impact on the scattering type generated by the Touzi decomposition is discussed in the following Sections 4.3 and 4.4.

### 4.3. PALSAR2 Recalibration: Impact on the Scattering Matrix Elements

The impact of the recalibration can be assessed through the comparison of the original scattering matrix $[S_\Omega] - orig$ (provided by JAXA in 2015) with the recalibrated scattering matrix $[S_\Omega]_{recal}$ generated using the actual transmit–receive PALSAR2 antenna's distortion matrices.

For a better understanding of the impact of the recalibration on $[S_\Omega]$, the following expression can be derived, under the assumption that the PALSAR2 antenna cross talk (lower than −40 dB) can be ignored:

$$\begin{bmatrix} S^\Omega_{HH-recal} \\ S^\Omega_{HV-recal} \\ S^\Omega_{VH-recal} \\ S^\Omega_{VV-recal} \end{bmatrix} \simeq \begin{bmatrix} S^\Omega_{HH-orig} + \delta_4.S^\Omega_{HV-orig} + \delta_2.S^\Omega_{VH-orig} \\ \delta_3.S^\Omega_{HH-orig} + S^\Omega_{HV-orig} + \delta_2.S^\Omega_{VV-orig} \\ \delta_1.S^\Omega_{HH-orig} + S^\Omega_{VH-orig} + \delta_4.S^\Omega_{VV-orig} \\ \delta_1.S^\Omega_{HV-orig} + \delta_3.S^\Omega_{VH-orig} + S^\Omega_{VV-orig} \end{bmatrix} \tag{7}$$

Equation (7) above expresses the four elements of the recalibrated scattering matrix $[S_\Omega]_{recal}$ (before the Faraday rotation correction) as a function of the original PALSAR2 scattering matrix $[S_\Omega] - orig$. Since L-band PALSAR2 cross-pol (HV and VH) are very low (10 dB) compared to the co-pol (HH and VV) in peatlands regions [21], and the antenna cross talks of Table 1 adopted in the original image are lower than −32 dB, HH and VV should not be affected (in magnitude and phase) by HV and VH, according to the expressions of $S^\Omega_{HH-recal}$ and $S^\Omega_{VV-recal}$ given the equation above (row 1 and row 4). In contrast to the co-pol (HH and VV), the low HV and VH should be significantly affected by the co-pol HH and VV, according to the expressions of $S^\Omega_{HV-recal}$ and $S^\Omega_{VH-recal}$ given in Equation (7) above (rows 3 and 4). These results are confirmed in the following.

Figures 5 and 6 present for the co-pol (HH and VV), the normalized ratio [46] (normalized ratio R: R is between 0 and 1, and the inverse of the ratio (1/R) is adopted if R is larger than 1) of the magnitude of the original PALSAR2 image and that of the recalibrated image, $|S^\Omega_{hh-orig}/S^\Omega_{hh-recal}|$ and $|S^\Omega_{vv-orig}/S^\Omega_{vv-recal}|$ (before the Faraday rotation correction). As expected, the HH and VV magnitudes as well as their phase differences in Figure 7 were

not affected by the original calibration, and the recalibration did not significantly change their values (less than 0.2 dB in radiometry and less than 5 degrees in phase).

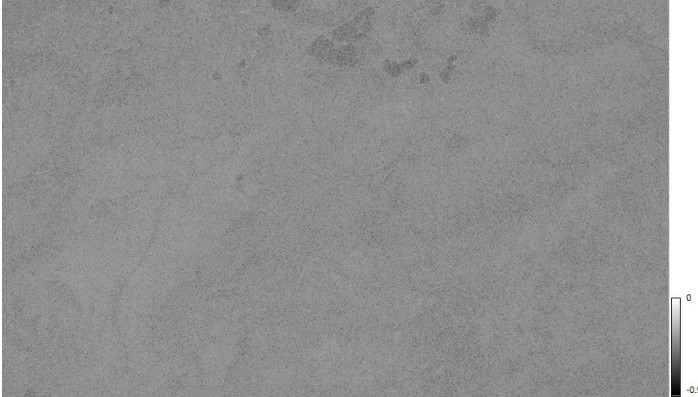

**Figure 5.** HH original–recalibrated magnitude ratio.

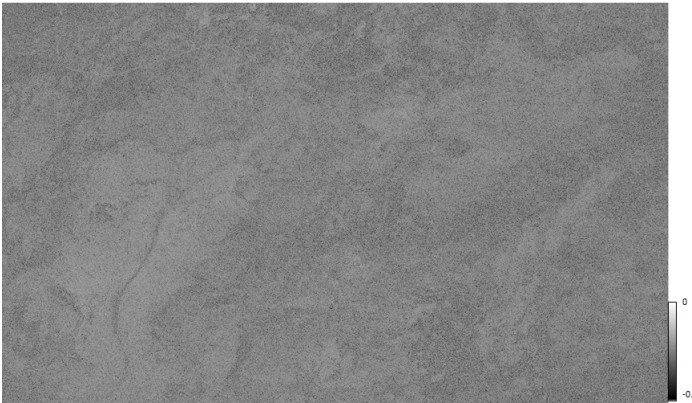

**Figure 6.** VV original–recalibrated magnitude ratio.

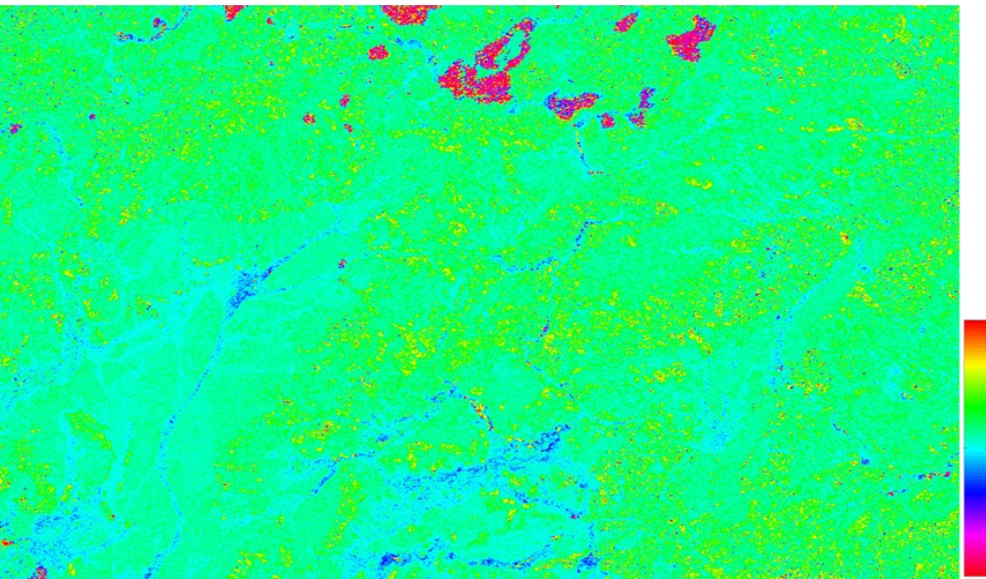

**Figure 7.** HH-VV phase difference.

Figures 8 and 9 present for the cross-polarization (HV and VH) the normalized ratio of the magnitude of the original PALSAR2 and the recalibrated image, $|S^{\Omega}_{hv-orig} / S^{\Omega}_{hv-recal}|$ and $|S^{\Omega}_{vh-orig} / S^{\Omega}_{vh-recal}|$ (before the Faraday rotation correction). In contrast to the co-pol, the cross-polarizations (HV and VH) of much lower intensity (than the co-pol) were

significantly affected (more than 1 dB) by the HH and VV cross talk contribution through $\delta_1$ and $\delta_2$, as seen in Figures 8 and 9. VH was significantly more affected than HV. VH was mainly affected by the HH contamination that resulted from the most significant antenna cross talk $\delta_1$ (Table 1 in dB). HV was mainly affected by VV through $\delta_2$, which was 1.6 dB lower than $\delta_1$. VH intensity is presented in Figure 10. The areas of low VH intensity values in Figure 10 correspond to a significant improvement (more than 1 dB) of VH, as can be seen in Figure 9. These areas (of low VH and HV intensity values), which are mainly assigned to the permafrost class (in pink colour) in Figure 1 and the treed bog class by the AWI and/or AGCC classifications of Figures 2 and 3, had the cross-pol (HV and VH) measures significantly affected by the like-pol (HH and VV) because of the miscellaneous errors on the antenna cross talks $\delta_1$ and $\delta_2$ correction. The PALSAR2 recalibration permits the removal of the HH and VV contamination from VH and HV measurements, and this led to a residual error ($-43$ dB [39]) that was insignificant with reference to the PALSAR2 NESZ ($-37$ dB).

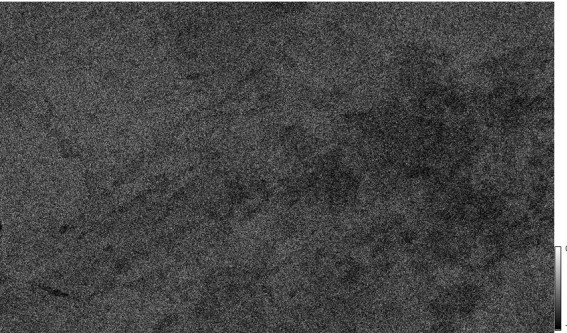

**Figure 8.** HV original–recalibrated magnitude ratio.

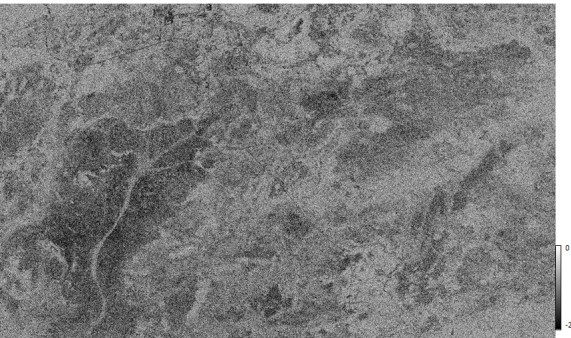

**Figure 9.** VH original–recalibrated magnitude ratio.

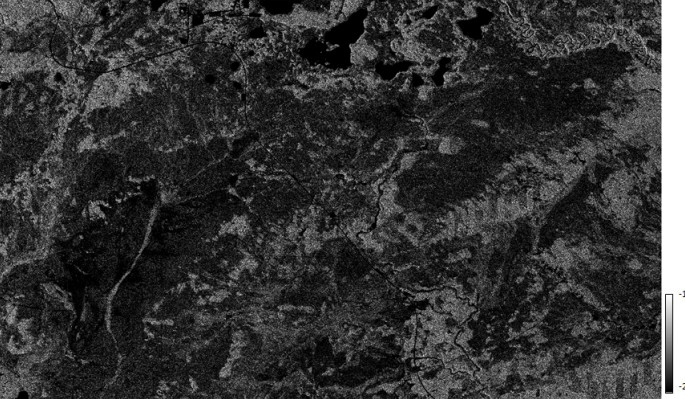

**Figure 10.** VH intensity in dB.

In the following, the Touzi decomposition is applied to the original and recalibrated images after the correction of the Faraday rotation contamination. It is worth noting that the Faraday rotation angle measure ($\Omega = 1.8°$) was not affected by the original calibration. Forested areas of relatively large cross-pol were used to measure $\Omega$ using the Bickel method [39,47,48], and this led to a measure that was not affected by the recalibration. In the following, the results obtained with the Touzi decomposition parameters derived using the original and recalibrated PALSAR2 are described. It is shown that the recalibration of PALSAR2 (with a residual error lower than $-43$ dB) permits a full exploitation of the excellent PALSAR2 NESZ (better than $-37$ dB) for the detection of relatively deep (up to 50 cm) discontinuous permafrost underlying peatlands [27,49,50].

### 4.4. PALSAR2 Image Recalibration: Impact on the Touzi Decomposition Main Parameters

The dominant-scattering-type phase opposite, $\phi_{s1o}^{orig}$ and $\phi_{s1o}$, derived from the original and recalibrated PALSAR2 images (after the correction of the Faraday rotation contamination) are presented in Figures 11 and 12, respectively. The $\phi_{s1o}^{orig}$ and $\phi_{s1o}$ images of Figures 11 and 12 are updated with the permafrost class contours of the AGS classification of Figure 1 to enhance the added value of the PALSAR2 recalibration in discontinuous permafrost areas. The absolute phase difference between $\phi_{s1o}^{orig}$ and $\phi_{s1o}$ is presented in Figure 13.

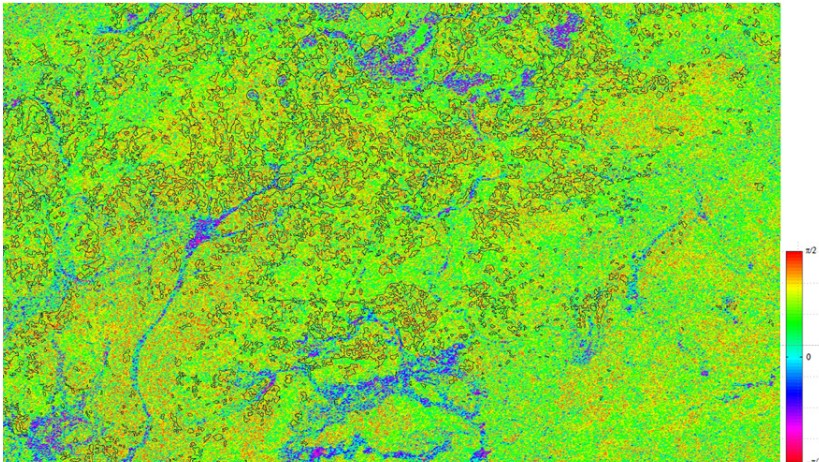

**Figure 11.** Scattering type Phis1o derived from the original ALOS2 image. The contours of permafrost areas generated from the AGS classification are included in the image.

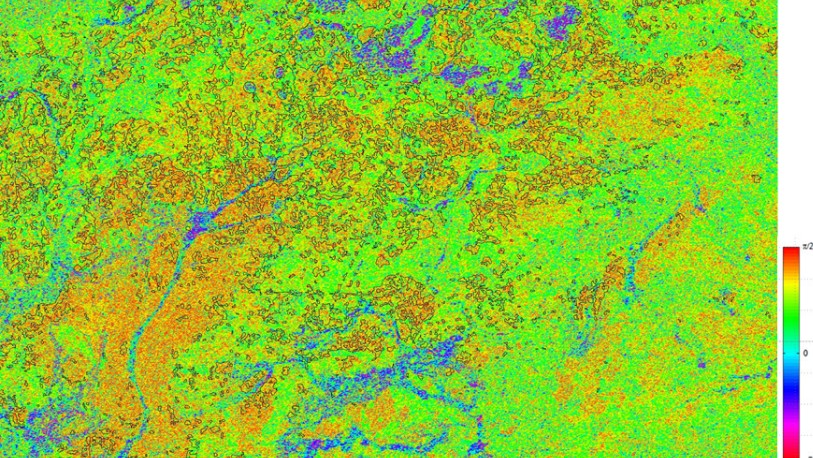

**Figure 12.** Scattering type phase Phis1o derived using the recalibrated ALOS2 image. The contours of permafrost areas generated from the AGS classification are included in the image.

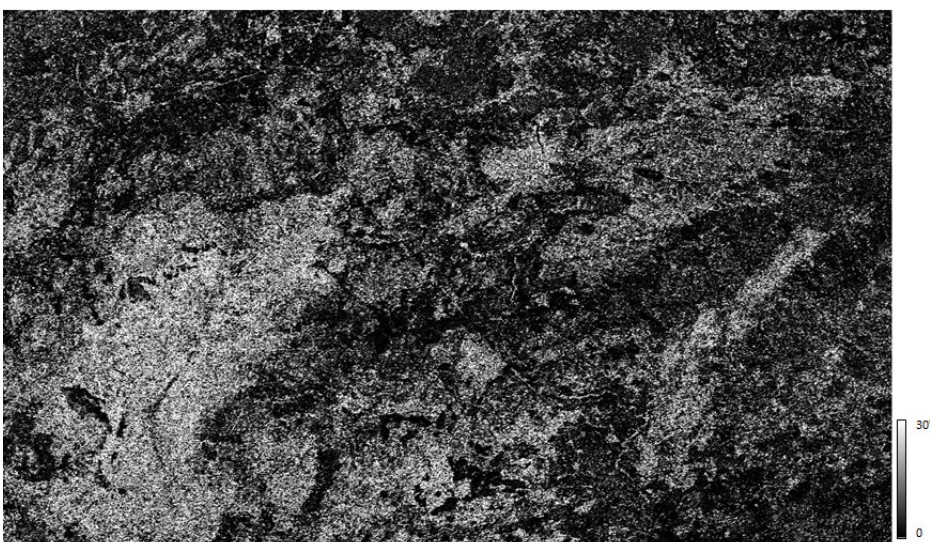

**Figure 13.** Phis1o recalibrated–original image error (absolute phase difference in degrees).

As expected, the recalibration led to the correction of a significant error (about 30 degrees), as seen in Figure 13. The latter presents the absolute phase difference $|\phi_{s1o}^{orig} - \phi_{s1o}|$ between the phase derived from the original and recalibrated PALSAR2 image. The areas of largest phase error corresponded to the ones of the largest VH error of Figure 9, as expected. These areas (of low VH and HV), which were mainly assigned to the permafrost class (in pink colour) in Figure 1 and the bog class by the AWI classification of Figure 3, had the cross-pol (HV and VH) measures significantly affected by the like-pol (HH and VV), and this led to a significant error on the dominant scattering type phase. The investigation conducted in [51] on peatland classification using the dominant scattering phase $\phi_{s1}^{orig}$ led the conclusion that the dominant-scattering-type phase $\phi_{s1}$ derived from the recalibrated FP6-4 images had to be used for an enhanced peatland characterization. This result was confirmed herein for a discontinuous permafrost mapping in peatland regions. The comparison of the phase error of Figure 13 and the recalibrated $\phi_{s1o}$ image of Figure 12 with the AGS classification of Figure 1 highlighted the significant enhancement of the scattering type phase $\phi_{s1o}$ for the enhanced identification of the discontinuous permafrost class (presented in orange in the scattering-type phase ($\phi_{s1o}$ and $\phi_{s1o}^{orig}$) images). The peatland subsurface permafrost areas brought out by the scattering-type phase $\phi_{s1o}$ (orange in Figure 13) generated using the recalibrated PALSAR2 images looked very similar to the permafrost class (pink in Figure 1) identified using the AGS LiDAR-Landsat classification. These results are confirmed in Section 5 using field data measurements collected by AGS during the ALOS2 data collection.

The medium scattering type phase $\phi_{s2}$ regenerated using the recalibrated PALSAR2 images is presented in Figure 14, and the absolute phase difference $|\phi_{s2}^{orig} - \phi_{s2}|$ between the phase derived from the original and recalibrated PALSAR2 images is presented in Figure 15. The recalibration of the medium-scattering-type phase $\phi_{s2}$ also permits a significant enhancement of the phase (about 20 degrees). This significant error can have a negative impact on the potential of $\phi_{s2}$ for discontinuous permafrost mapping, as confirmed in the detailed investigation conducted in Section 5.

It is worth noting that the scattering-type magnitude $\alpha_{s1}$ of Figure 16, which is not sensitive to subsurface discontinuous permafrost, was not significantly improved by the recalibration, less than 5° after the PALSAR2 image recalibration, as seen in Figure 17. Notice that the scattering-type magnitude $\alpha_s$ was already shown to be insensitive to peatland subsurface water in [21]. It was also shown that the very promising results obtained with PALSAR for peatland subsurface water flow monitoring could not be obtained if the PALSAR NESZ was not designed with the excellent NESZ of −34 dB [20–22,29].

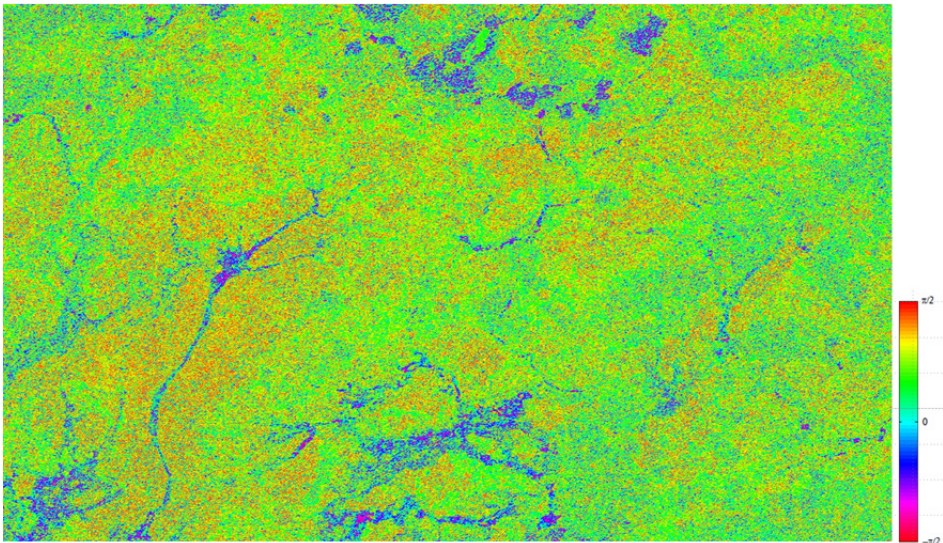

**Figure 14.** Scattering-type phase Phis2 derived using the recalibrated ALOS2 image.

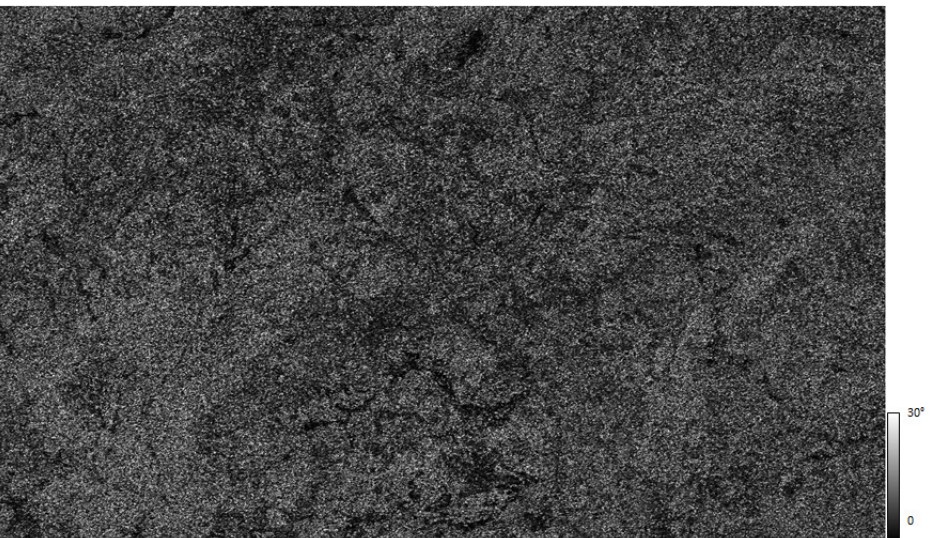

**Figure 15.** Phis2 recalibrated–original image error (dB).

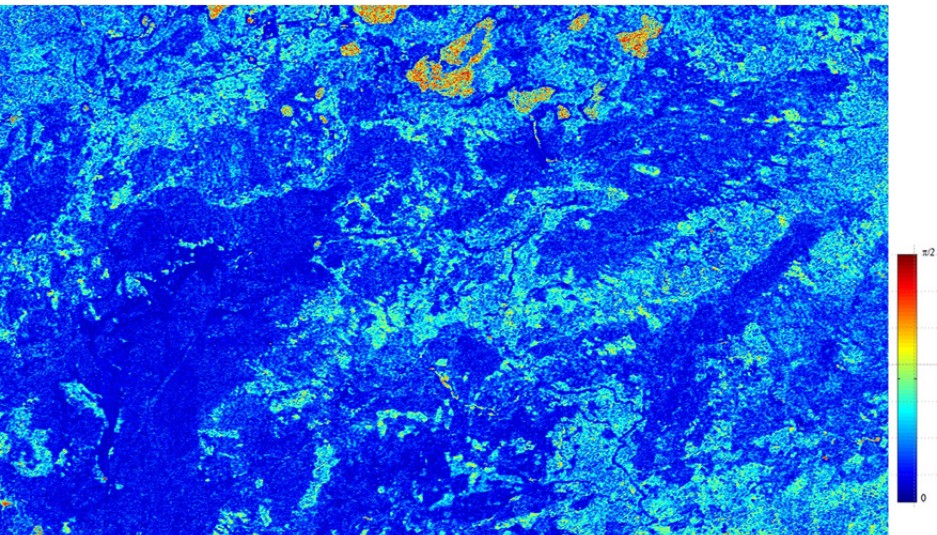

**Figure 16.** Dominant-scattering-type magnitude alphas1 obtained using the recalibrated ALOS2 image.

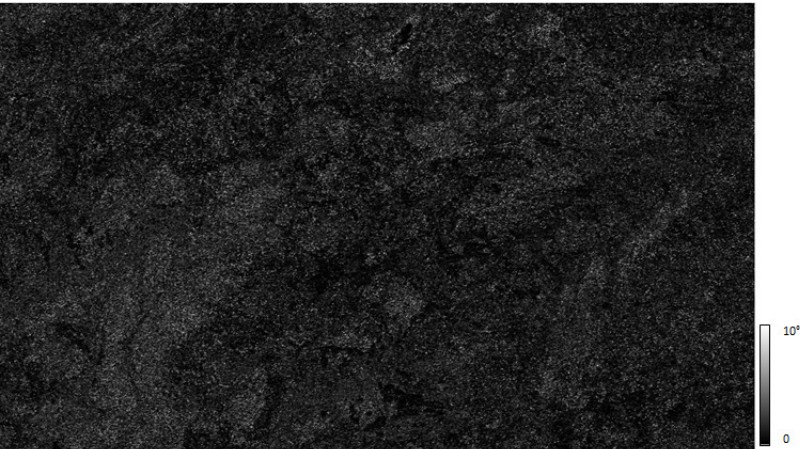

**Figure 17.** alphas1 calibration error: absolute difference (in degrees) between the scattering-type magnitude obtained from the original image and the one obtained from the recalibrated image.

In summary, the recalibration of PALSAR2 permitted a significant improvement of the dominant- and medium-scattering-type phases $\phi_{s1o}$ and $\phi_{s2}$. The correction of the significant residual error generated by the miscellaneous problem that occurred with the FP6-4 beam calibration should permit a full exploitation of the excellent NESZ ($-37$ dB) of PALSAR2 for the enhanced detection of subsurface discontinuous permafrost areas in peatland regions, as demonstrated in Section 5.

Notice that the miscellaneous problem with the calibration of the PALSAR2 FP6-4 beam mode was recently solved by JAXA, and an updated list of the polarimetric calibration parameters (transmitter–receiver distortion matrix, as well as channel imbalances) that had been used since January 2018 was provided [52].

## 5. Investigation of Recalibrated Polarimetric ALOS2 for Enhanced Discontinuous Permafrost Mapping

### 5.1. Polarimetric PALSAR2 Image Analysis

A PALSAR2 polarimetric information optimization was conducted through the application of an incoherent target decomposition (ICTD) on the recalibrated polarimetric ALOS2 data collected with the highly sensitive PLR mode of about 4.4 m × 5.1 m resolution at FP6-4 (about 27 degrees of incidence angle). The Touzi decomposition was applied using an 11 × 11 moving window. The moving window included more than 80 independent samples, and this permitted meeting the requirement setup in [15] for an unbiased estimation of the Touzi decomposition. The Cloude–Pottier ICTD main parameters $\alpha$ and H were also derived using an 11 × 11 moving window. The extrema of the degree of polarization, the so-named Touzi discriminators [21,53,54], were also generated to exploit the important complementary information they can provide for an enhanced peatland classification. In contrast to the ICTD, a smaller moving window (7 × 7) can be used to provide unbiased estimates of the DoP extrema, as shown in [21,54]. The minimum degree of polarization (DoP), pmin, was shown to be among the key polarimetric parameters that permitted an enhanced discrimination of treed bog from upland forests in the Wabasca peatland region (Northern Alberta) [21]. In addition to all these polarimetric parameters cited above and to the conventional multipolarization (HH, HV, and VV) channels, the HH-VV phase difference ($\phi_{HH}$-$\phi_{VV}$) was also investigated. The HH-VV phase difference, which used to be among the standard polarimetric parameters investigated for a natural target characterization [55–57], has been widely promoted for the detection of sand subsurface wet structures in arid regions [58,59].

### 5.2. Field Data and Study Area Investigated

In conjunction with JAXA, a polarimetric ALOS2 acquisition was planned, at the maximum permafrost thaw conditions, on the 23rd of August at beam FP6-4 (27° incidence

angle). Field data were collected during the week of the ALOS2 acquisition over discontinuous permafrost distributed amongst peat plateaus and wooded palsa bogs near Namur Lake. The purpose of the field data collection was to characterize ground cover, wetland type, peat depth and permafrost depth (if present). Complementary information related to geomorphology, relief, lithology, and soil moisture was also collected during the 2014 field work. The individual study sites included areas with and without permafrost, due to the discontinuous nature of the permafrost in the area. Most permafrost sites were found to be in bog regions dominated by black spruce and various depths of peat. Ground cover at these permafrost sites was typically a mixture of peat mosses (Sphagnum), Labrador tea (Rhododendron groenlandicum), and caribou lichen (Cladonia rangiferina). At sites where permafrost was not found, a mixture of wetland types and forest types was found. Common wetland types at these sites include bogs (open, treed), fens (treed, shrubby, open), and swamps. Forests in the area are common and mixed, with a variety of conifer and deciduous species of varying density found in the area. The most common dominant tree species include black spruce (Picea mariana), white spruce (Picea glauca) and pine species (Pinus).

### 5.3. ALOS2-PALSAR2 Image Analysis

In the following, more focus is assigned to the area outlined with a rectangle on Figure 4. The multipolarization (HH, HV, VV) PALSAR2 image of the area of interest is presented in Figure 18. All the field-sample locations are presented in the PALSAR2 multipolarization image of Figure 18. The image scale, the four cardinal directions (N, S, E, W), and the coordinates that mark the geolocation are also indicated in Figure 4. The corresponding AGS, AWI, and AGCC classifications are presented in Figures 19–21. The Touzi decomposition was applied to the study area. Figures 22 and 23 present the dominant- and medium-scattering-type phases $\phi_{s1o}$ and $\phi_{s2}$. The phase difference of HH and VV is presented in Figure 24.

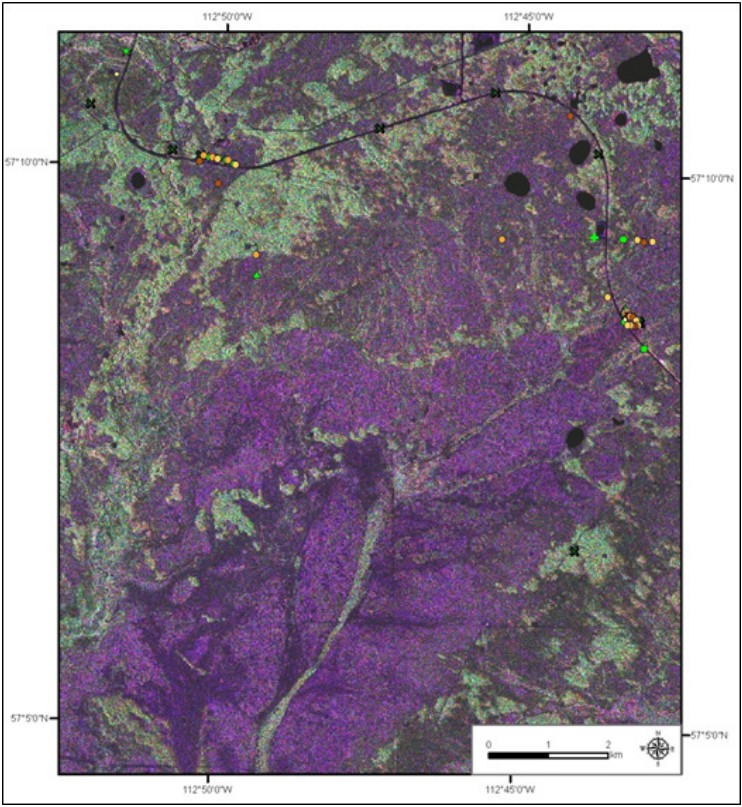

**Figure 18.** Study area: multi-pol ALOS2 image.

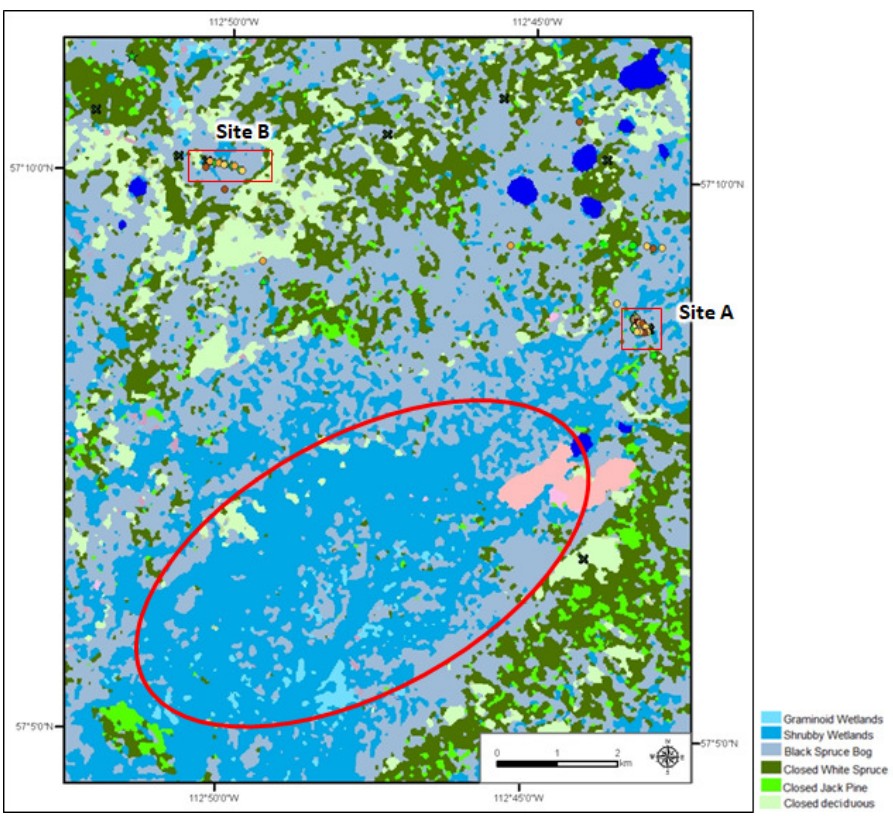

**Figure 19.** AGS classification.

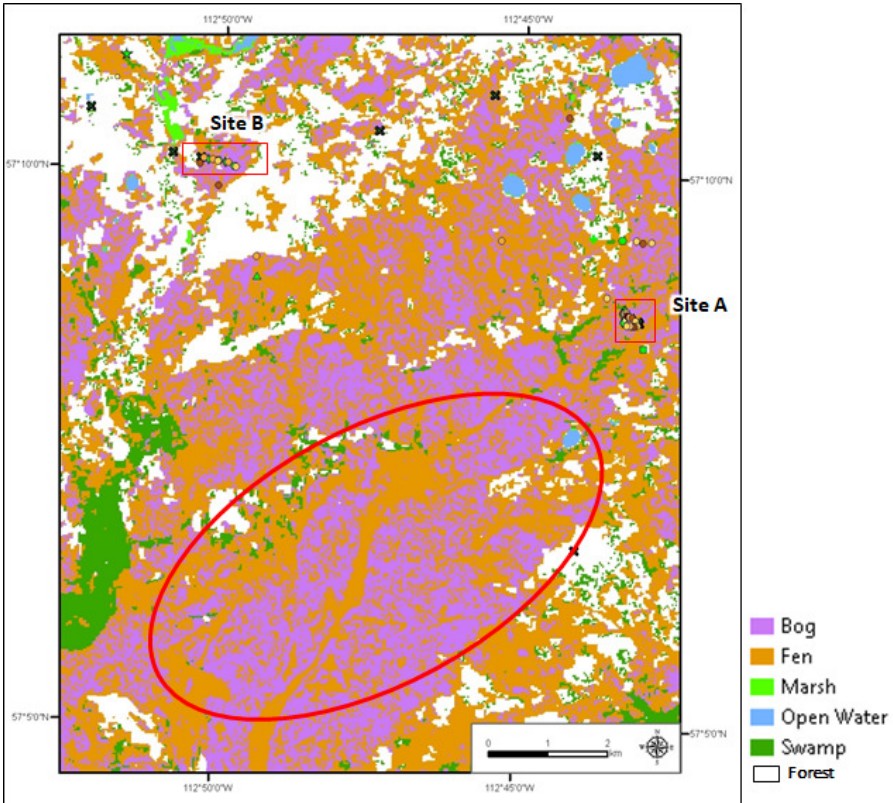

**Figure 20.** AWI classification.

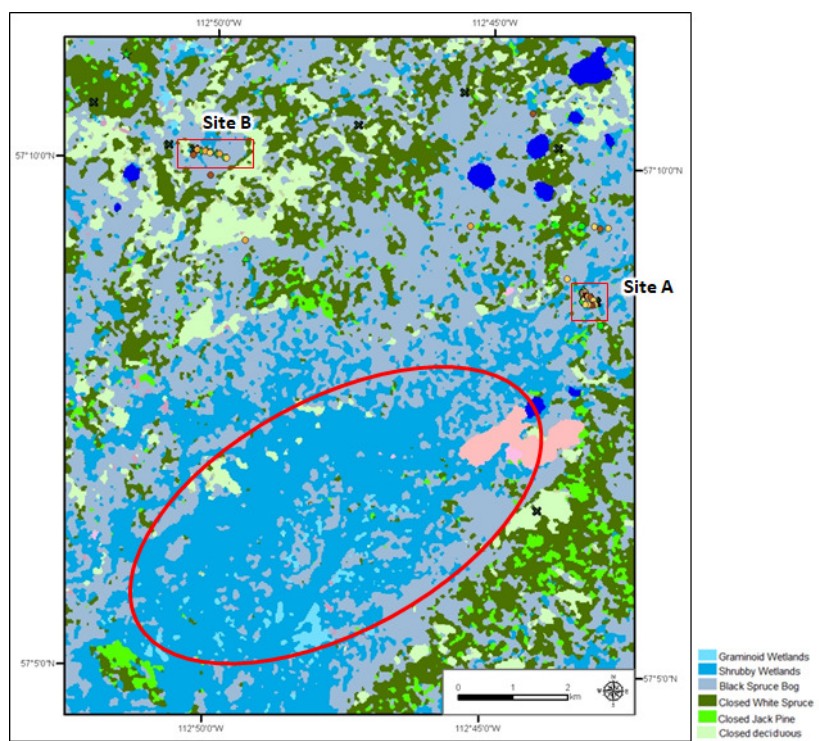

**Figure 21.** AGCC classification: burned areas are presented in light pink.

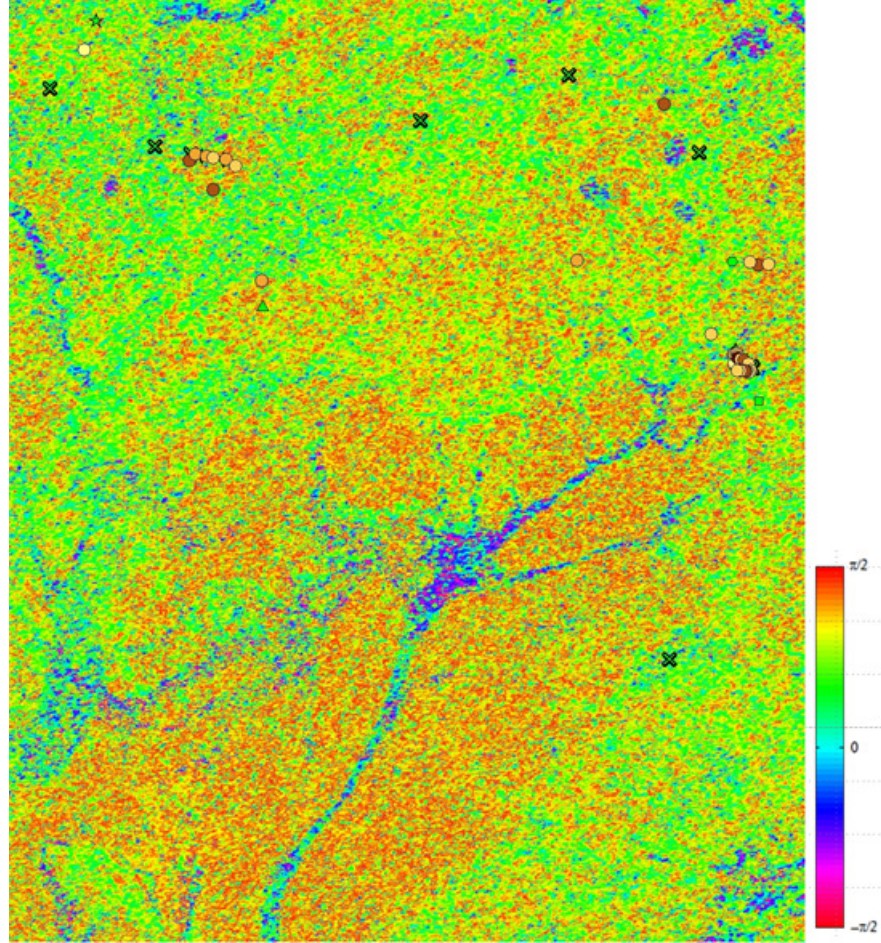

**Figure 22.** Touzi dominant-scattering-type phase Phis1o derived using the recalibrated ALOS2 image.

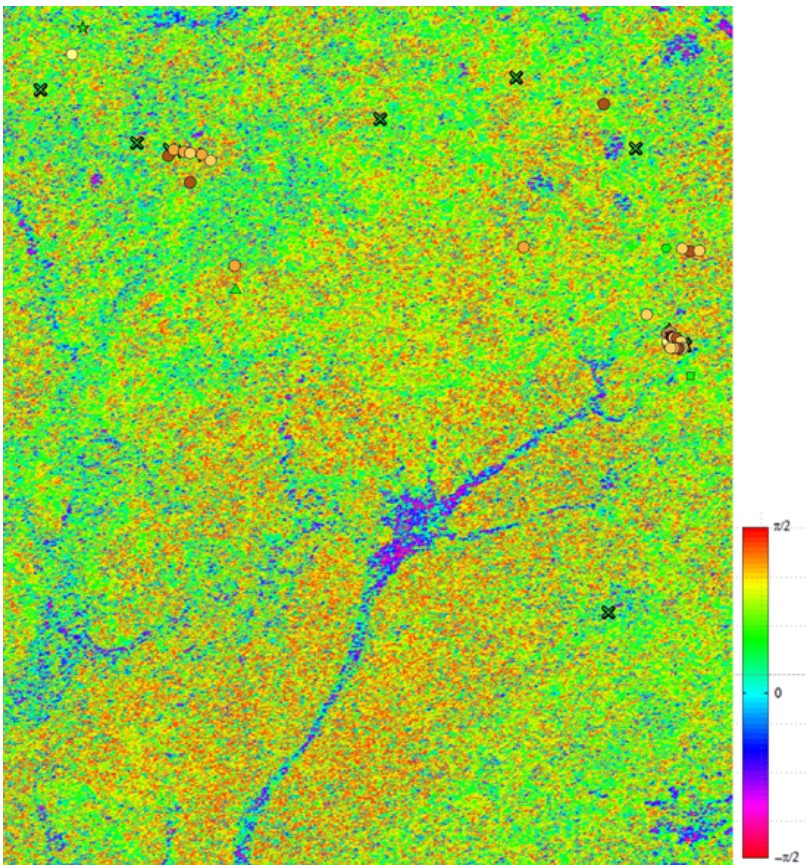

**Figure 23.** Touzi medium-scattering-type phase Phis2 derived using the recalibrated ALOS2 image.

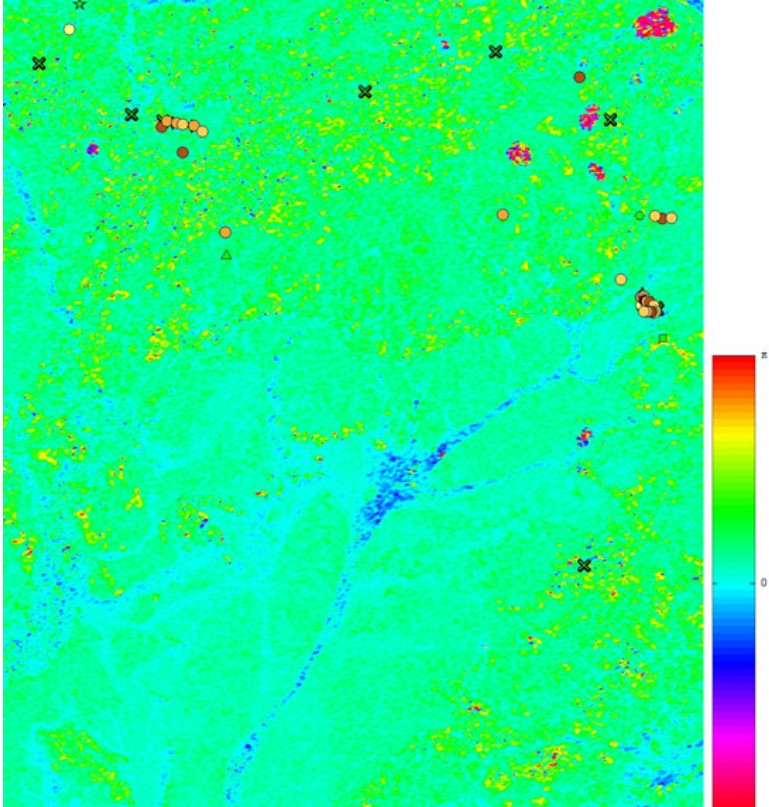

**Figure 24.** HH-VV phase difference.

A comparison between the AWI and Landsat-based AGCC wetland classifications of Figures 20 and 21 revealed a large difference between the bog-fen classes. A large area (in pink) assigned to the bog (treed and open bogs) class (outlined in Figures 20 and 21) was assigned by the AGCC to the shrubby wetland class (blue) in Figure 21. In fact, similar AGCC bog misclassification was previously brought out in [21] on the Wabasca study area. It was shown that the minimum DoP, pmin, generated from polarimetric ALOS data permitted an enhanced discrimination of treed bogs from upland forests [21]. The combination of pmin and pmax with the Landsat-based AGCC classification led to a treed bog class similar to the one obtained by the AWI [21], as confirmed herein at the Namur Lake study site. Figures 25 and 26 present the minimum and maximum DoP, pmin and pmax, generated on the Namur Lake using the recalibrated ALOS2 image. The analysis of the pmin image of Figure 25 with reference to the AWI of Figure 20 confirmed the results obtained in [21]. The large area, which was assigned to the treed bog by the AWI, had a relatively large value of pmin (larger than 0.3), a very high pmax value (higher than 0.8), and looks bright in Figure 25. As in our previous investigation on the Wabasca area [21], the treed bogs were misclassified by the Landsat-based AGCC classification. These treed bogs have been accurately identified by the AWI classification at the Namur lake study site. The AWI, which is based on high-resolution aerial photographs, provided more details in comparison with the 30 m Landsat-based AGCC classification. As a result, the AWI led to a more accurate treed bog class identification in comparison with that of the AGCC, as shown in [21] and confirmed herein. The treed-bog area outlined in Figures 20 and 21 was also misclassified by the LiDAR-Landsat AGS classification of Figure 19 and assigned to the fen class. Since the AGS LiDAR-Landsat-based classification assumes that the permafrost is located in bog plateaus (under hummock) with extensive caribou lichen, the treed-bog area also outlined on Figure 19 was misassigned by the AGS to the nonpermafrost class, as discussed in Section 5.3.3.

For an effective investigation of the added value of the polarimetric PALSAR2 information (optimized with the Touzi decomposition) in support of an enhanced discontinuous permafrost mapping, more focus was assigned herein on the two sites (site A and site B) outlined in Figure 18 over which a large number of samples was investigated during the 2014 field data collection. The sites A and B are also outlined on the AGS permafrost classification and the AWI and AGCC wetland classifications of Figures 19–21. The detailed analysis of the results obtained with PALSAR2 on these two sites permitted the demonstration of the key information provided by Touzi dominant- and medium-scattering-type phases for the accurate detection of subsurface discontinuous permafrost, in comparison with the other parameters generated by the Touzi decomposition, the Cloude–Pottier $\alpha$-H ICTDs, the HH-VV phase difference, and the multipolarization (HH, HV, and VV) channels. The performance of the dominant- and medium-scattering-type phases, $\phi_{s1}$ and $\phi_{s2}$, were also compared using the field data collected by the AGS during the ALOS2 image acquisition.

### 5.3.1. ALOS2 Results: Site A

Figures 27–29 present the AGS permafrost classification and the AWI and AGCC wetland classifications. The three classifications indicate the field-sample identifications (a number) and locations. The AGS permafrost classification of Figure 27 indicates in addition the permafrost or nonpermafrost class each sample is assigned to. The sites underlain by subsurface permafrost are presented with circles filled with a colour that indicates the permafrost depth (the colour identifies the depth (in meters) interval among five intervals (0.15–0.30, 031–0.50, 0.51–0.85, 0.86–1.20, 1.21–1.80). The samples not underlain by permafrost are identified in Figures 27 by markers (in green) with the shape indicating the class they belong too. The AGS permafrost class each sample is assigned to and the depth of the permafrost are indicated in Table 2.

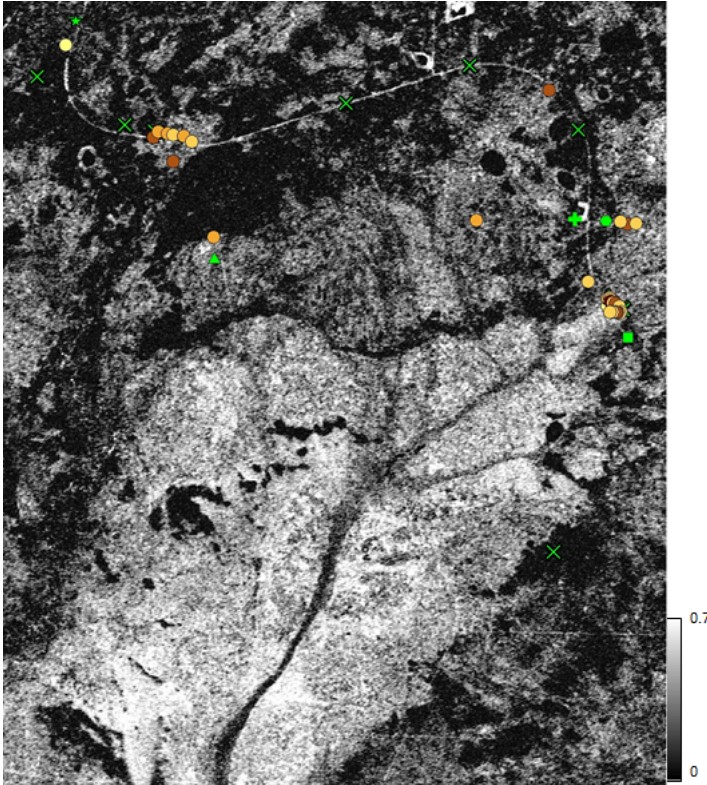

**Figure 25.** Minimum DoP pmin.

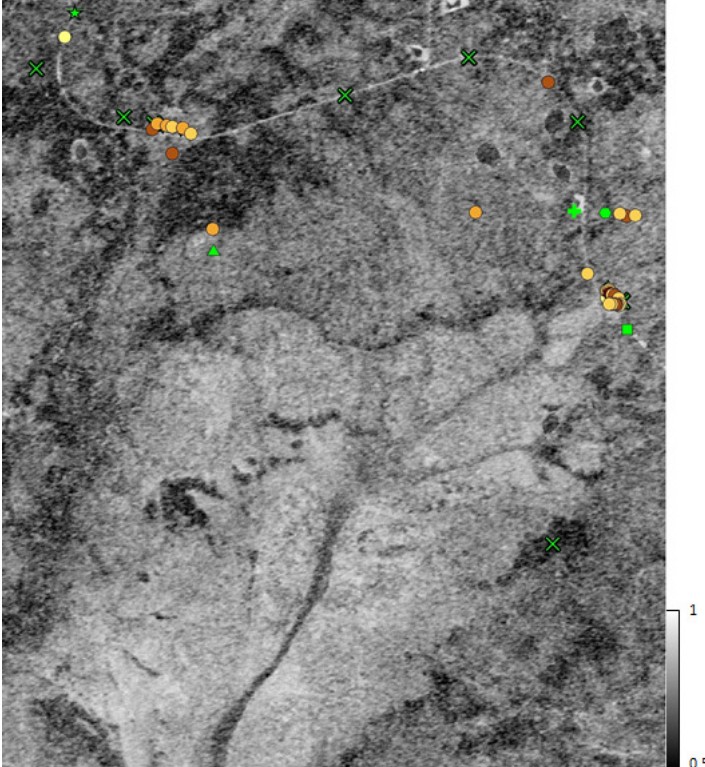

**Figure 26.** Maximum DoP pmax.

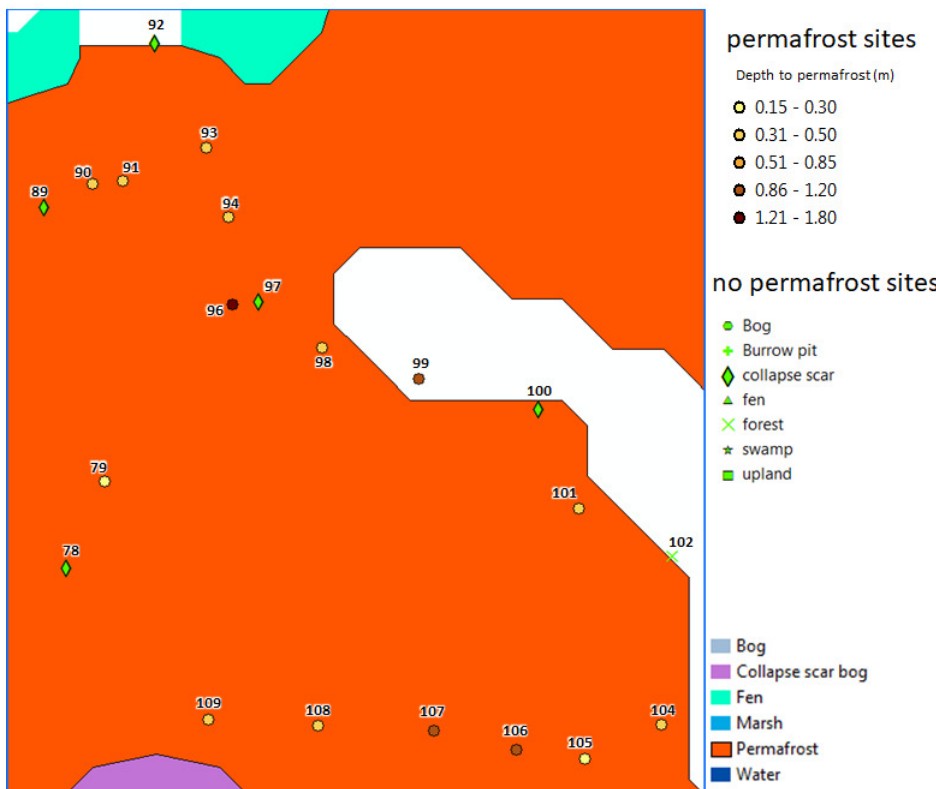

**Figure 27.** Site A: AGS permafrost classification.

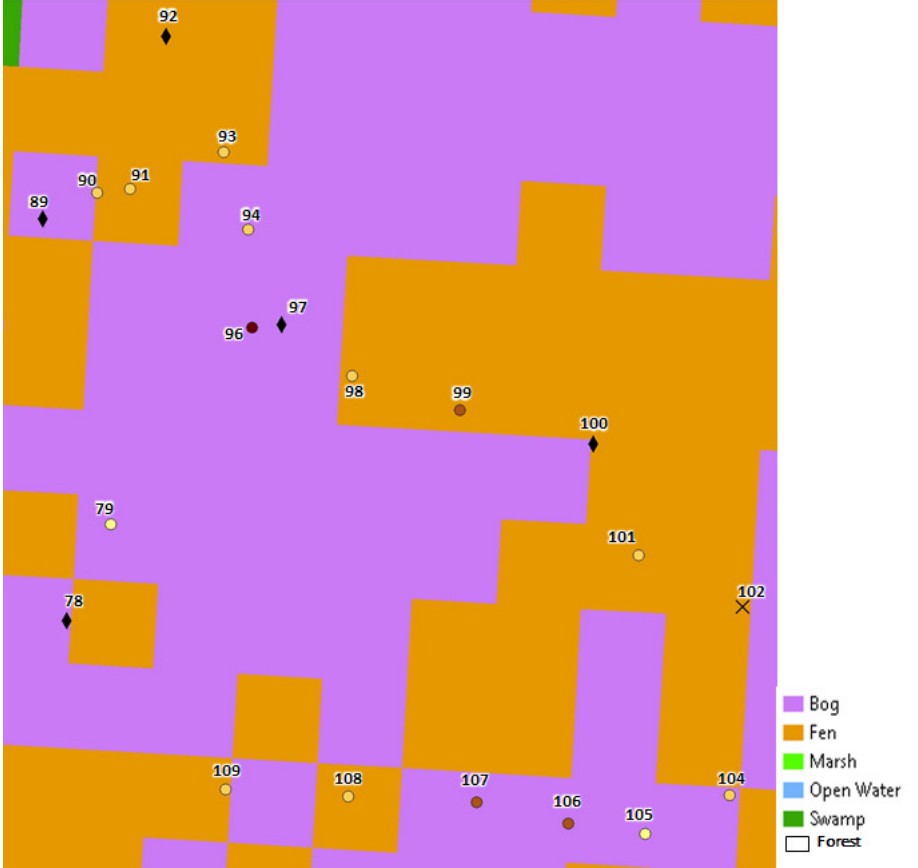

**Figure 28.** Site A: AWI classification.

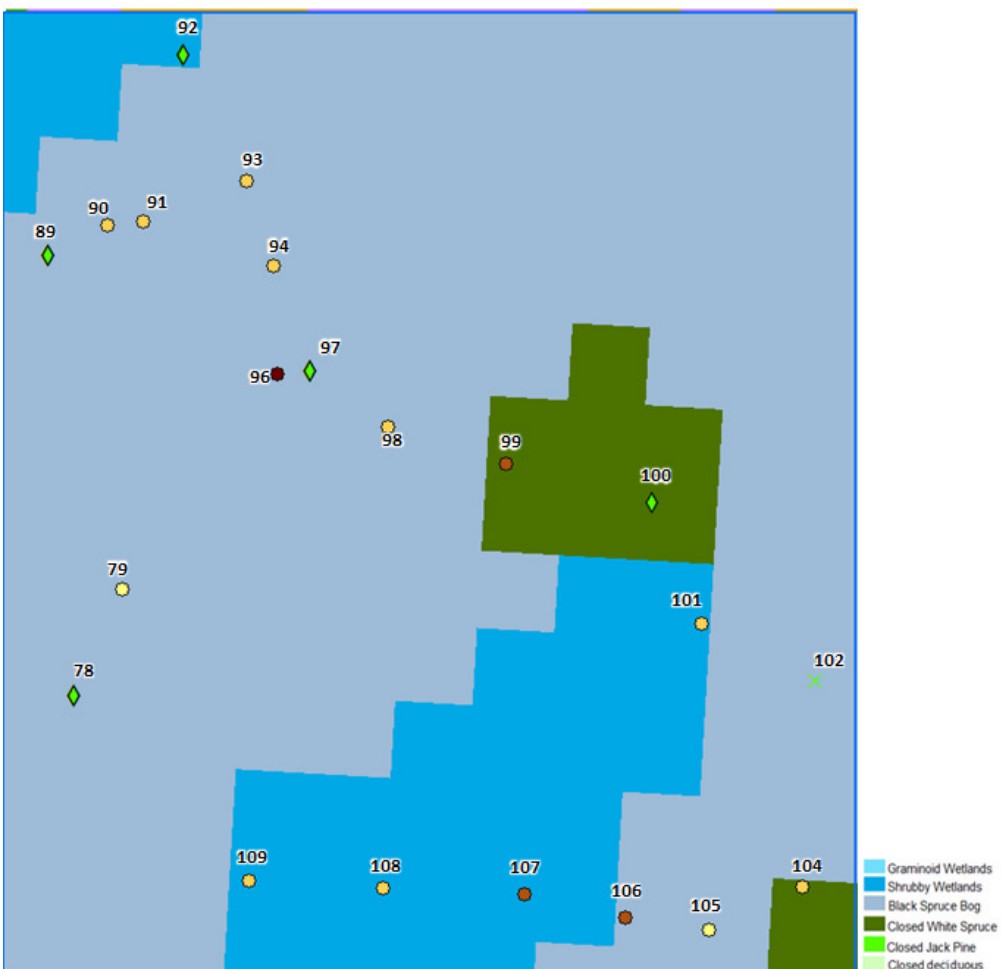

**Figure 29.** Site A: AGCC classification.

A global analysis of the AWI and AGCC wetland classifications of Figures 28 and 29 indicated that the site was dominated by the treed bog class. The analysis of the AGS permafrost classification indicated that most of the area was assigned to the permafrost class (pink in Figure 27). It is worth noting that the AGS LiDAR-Landsat permafrost classification of Figure 27 assumed that discontinuous permafrost was located in bog plateaus (under hummock) with extensive caribou lichen. LiDAR and Landsat data were combined to identify these areas, which were assigned to the permafrost class [5,6].

The detailed analysis of the AGS classification of Figure 27 using the field data collected revealed that few samples not underlain by permafrost, which belonged to the collapse scar (CS) class, (CS-78, CS-89, CS-92, CS-97, and CS-100) indicated by a green lozenge in Figure 27, were assigned to the permafrost class. The latter was supposed to include only the areas underlain by permafrost. All the remaining samples were assigned to the permafrost class with a colour that indicated the permafrost depth. The following questions could be brought out during the investigation of the added value of polarimetric ALOS2 imagery for an enhanced permafrost classification:

1. Is it possible for polarimetric ALOS2 imagery to identify all the permafrost samples and discriminate them from the nonpermafrost samples located in areas not underlain by permafrost?
2. Given the limited penetration of the L-band wavelength (much better than the C-band but still limited in comparison with the P-band), is it possible to identify accurately the permafrost samples?

3.  Is it possible to adjust the decision regarding deep versus very deep permafrost samples using tools that measure the reliability of the information provided by polarimetric ALOS2.

4.  What is the maximum depth at which the long penetrating polarimetric ALOS2 is sensitive to permafrost? How deep is the permafrost that can be detected?

One of the objective of the study conducted herein was to investigate the potential of polarimetric ALOS2 information for the exclusion of the sites not underlain by permafrost from the permafrost class. The second objective was to assess the maximum depth at which the long-penetrating polarimetric PALSAR2 was sensitive to, in the presence of permafrost. The dominant- and medium-scattering-type phases, $\phi_{s1}$ and $\phi_{s2}$, combined with the minimum DoP, pmin, and the Huynen maximum polarization return ($m$ given by Equation (2)) should provide the required information in support of an enhanced permafrost detection (up to 50 cm) and the identification of the discontinuous permafrost distribution using polarimetric ALOS2 data, as discussed in the following.

Figure 30 presents the dominant-scattering-type phase (opposite) $\phi_{s1o}$ and the medium-scattering-type phase $\phi_{s2}$. The phase of the target scattering type was shown to be sensitive to peatland subsurface water flow [21,39], and as such, we should expect very useful information from $\phi_{s1o}$ and $\phi_{s2}$ on peatland subsurface discontinuous permafrost characterization. The minimum of the DoP, pmin, and the medium scattering's maximum polarization return, $m_2$, are presented in Figures 31 and 32. The HH-VV phase difference image is presented in Figure 33. The curves that represent the multipolarization (HH, HV, VV) intensities and spans for the various sites are given in Figure 34. Those of $\phi_{s1o}$, $\phi_{s2}$, and the HH-VV phase difference are presented in Figure 35. The curves of extrema of the DoP, pmin and pmax< are given in Figure 36. The Touzi (dominant-, medium-, and low-)scattering-type magnitudes and the Cloude $\alpha$ are presented in Figure 37. The ICTD eigenvalues and Cloude entropy H are presented in Figure 38. The multipolarization (HH, HV, and VV) measures over the various sites and corresponding span, pmin, pmax, and HH-VV phase difference are given in Table 2. Table 3 presents the measures of $\phi_{s1o}$, $\phi_2$, the scattering-type magnitudes, the Cloude alpha, and the ICTD eigenvalues and entropy for the various sites. The AGC class each sample was assigned to and the depth of permafrost at the sample location are indicated in Table 2.

The comparison of $\phi_{s1o}$ and $\phi_{s2}$ of Figure 30 and the corresponding curves of Figure 35 using the field data and the information on the samples Table 2 identified in Figures 27–29 leads to the following points:

-   The dominant- and medium-scattering-type phases, $\phi_{s1o}$ and $\phi_{s2}$, identified the sites (bog permafrost (BF)) of "relatively" deep (up to 50 cm) permafrost (BF79-to-109 in Table 2) with a phase, presented in orange in Figure 30, of value between 68° and 85° according to Table 3 and Figure 35.
-   The comparison of $\phi_{s1o}$ and $\phi_{s2}$ revealed that $\phi_{s2}$ performed better than $\phi_{s1o}$.
-   All the permafrost sites (of depth up to 50 cm) (BF-79, BF-90, BF-91, BF-93, BF-94, BF-98,BF-101, BF-104, BF-108, and BF-109) were detected by $\phi_{s2}$. $\phi_{s1o}$ missed the bog permafrost sites (BF-79, BF-91, BF-93, BF-98, BF-108, and BF-109) with phase values outside the permafrost class range (between 68° and 85°) according to Table 3.
-   Site BF-105: This bog permafrost site was missed by both $\phi_{s2}$ and $\phi_{s1o}$, according to Table 3 and Figure 35. The BF-105 site was originally assigned to a treed bog underlain by a relatively deep (30 cm) permafrost, according to Figure 28 and Table 2. A detailed analysis of the field data collected at this site revealed that the site was not underlain by a thick layer of permafrost. Ice was only present as thin lenses within a very thin peat cover, rather than at many sites where a contiguous and thick layer of frozen peat was encountered. As a result, both $\phi_{s2}$ and $\phi_{s1o}$ produced values outside the phase range required by the permafrost class, $\phi_{s2} = 47°$ and $\phi_{s1o} = 63°$ according to Table 3. This result confirmed the reliability of the scattering-type phases, and $\phi_{s2}$ in particular, in the assignment of the samples not underlain by permafrost to the nonpermafrost class.

- Deep bog permafrost sample: The scattering-type phase generated from ALOS2 could not detect permafrost deeper than 50 cm. All the deep permafrost samples (DBF99, DBF106-107) were not assigned by $\phi_{s2}$ to the permafrost class, according to Table 3 and Figure 35.
- Very deep permafrost sample (VDF-CS-96): According to the field data collected by the AGS, the area is located in a treed bog dominated by collapse scar vegetation, with very deep permafrost (more than 1.8 m). The sample was assigned by $\phi_{s1} = 49°$ to bog. $\phi_{s2} = 86.14°$, which was slightly larger than the maximum permafrost range (85), did not assign it to the permafrost class either. The low value of the Huynen maximum polarization return $m_2$ confirmed the weak return from the very deep permafrost. In fact, the samples of very low $m_2$ outlined in Figure 32 should be excluded from the permafrost class prior to the consideration of the medium-scattering-type phase $\phi_{s2}$ information.
- Collapse scar (SC) sites: $\phi_{s2}$ measured over all the collapse scar sites (CS78, CS-89, CS-92, CS-97, and CS-100) and the forest conifer (FC) site (FC102) confirmed that all these samples, which were collected in areas not underlain by permafrost, were not assigned to the permafrost class, according to Table 3 and Figure 35.
- The information provided by pmin and pmax in Table 2 permitted excluding several sites from the permafrost class, DBF99, CS100, and FC102, with a very low pmin (pmin lower than 0.2). The low pmin value indicated that these areas were not located in peatlands, as demonstrated in [21].
- It is worth noting that the permafrost samples' detection and their discrimination from the nonpermafrost areas could not be realized using the multipolarization (HH, HV, VV) intensities, span, and the HH-VV phase differences, as seen in Tables 2 and 3 and Figures 34, 35 and 37.
- The ICTD eigenvalues, the Cloude entropy H, the (dominant-, medium-, and low-scattering-type magnitudes, and the Cloude $\alpha$ did not permit the identification of permafrost samples either, according to Tables 2 and 3 and Figures 37 and 38.

The results above were confirmed on site B, as discussed in the following.

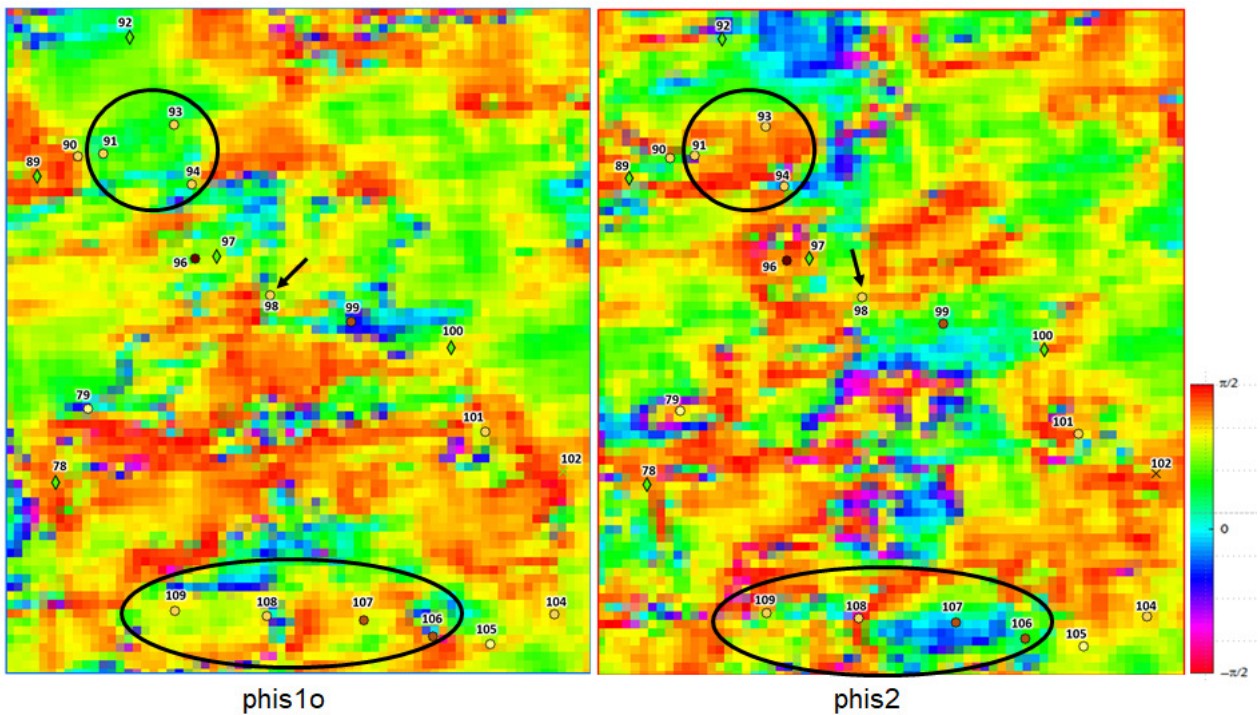

**Figure 30.** Site A: Touzi scattering-type phases: phis1o and phis2.

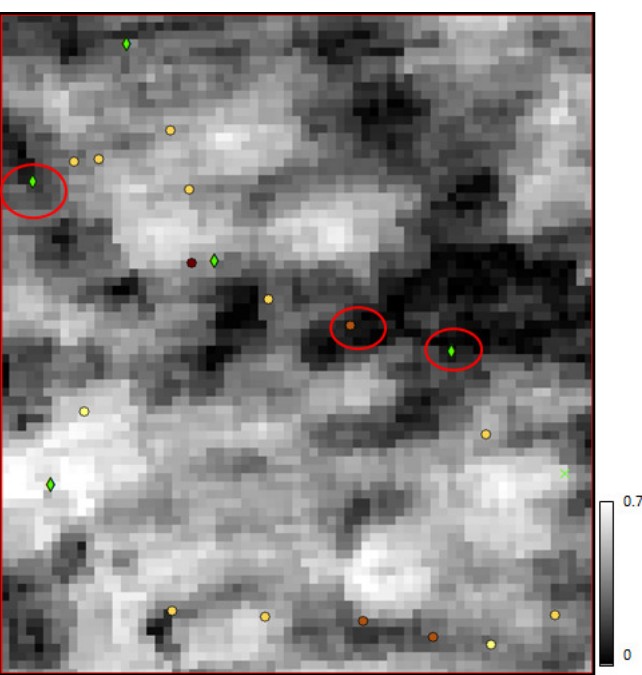

**Figure 31.** Site A: minimum DoP pmin. The legend of the different colored dots assigned to the permafrost and non-permafrost sites is given in Figure 27.

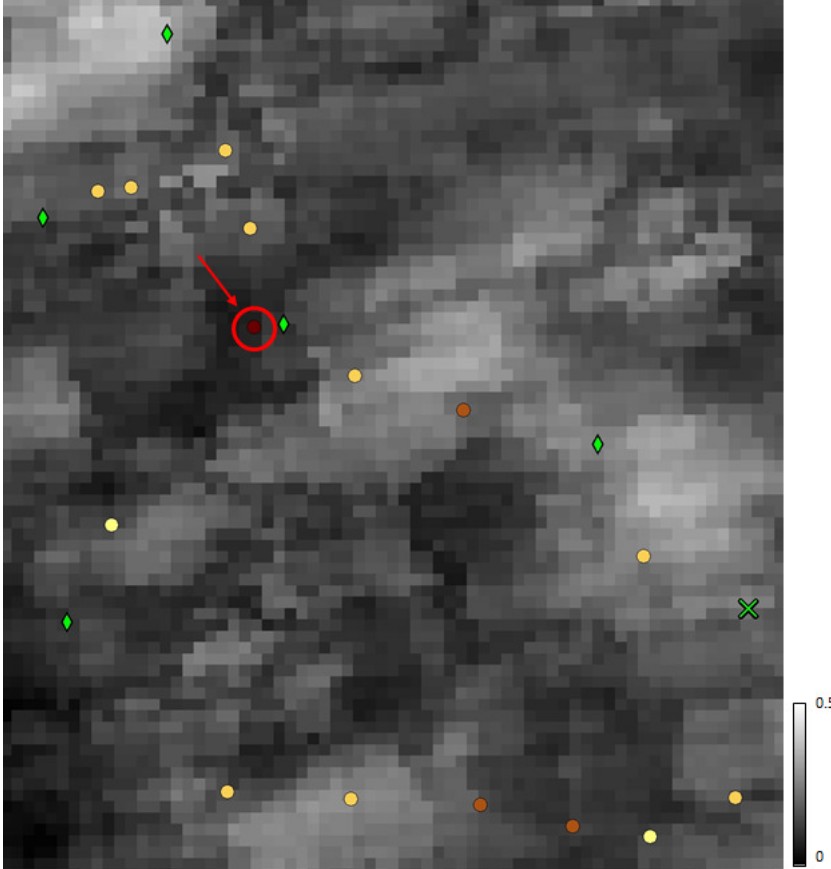

**Figure 32.** Site A: medium scattering's Huynen maximum polarization return ($m_2$). The legend of the different colored dots assigned to the permafrost and non-permafrost sites is given in Figure 27.

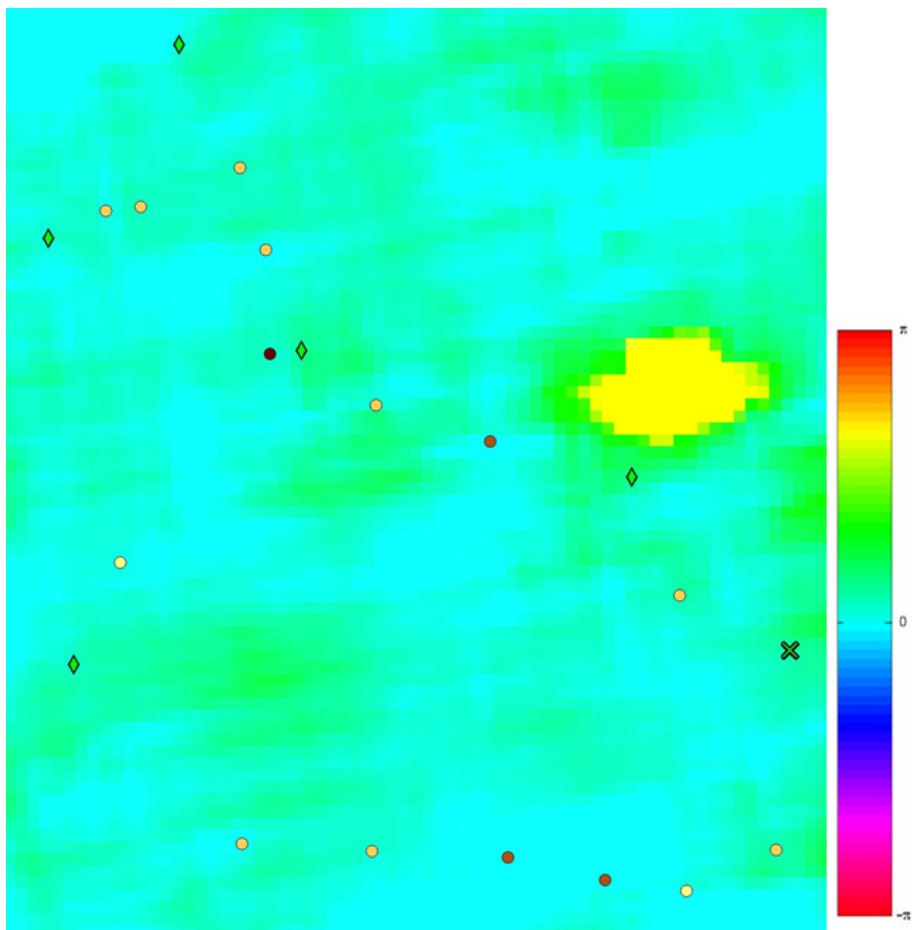

**Figure 33.** Site A: HH-VV phase diffrence. The legend of the different colored dots assigned to the permafrost and non-permafrost sites is given in Figure 27.

**Table 2.** Sample Class, Permafrost depth, and Conventional Polarimetric Parameters.

| Site ID | Site Type | ALT(m) | $HH_{dB}$ | $HV_{dB}$ | $VV_{dB}$ | Span | $\phi_{HH} - \phi_{VV}$ | pmin | pmax |
|---------|-----------|--------|-----------|-----------|-----------|------|-------------------------|------|------|
| BF-79 | Bog Permafrost | 0.3 | −10.35 | −19.44 | −10.55 | −6.92 | -24.34 | 0.54 | 0.87 |
| BF-90 | Bog Permafrost | 0.5 | −10.20 | −16.94 | −9.91 | −6.23 | -26.88 | 0.46 | 0.78 |
| BF-91 | Bog Permafrost | 0.4 | −9.42 | −17.97 | −9.66 | −5.94 | −21.43 | 0.35 | 0.78 |
| BF-93 | Bog Permafrost | 0.4 | −8.93 | −18.87 | −9.40 | −5.70 | −32.63 | 0.49 | 0.86 |
| BF-94 | Bog Permafrost | 0.5 | −9.68 | −18.46 | −9.53 | −6.07 | −34.53 | 0.44 | 0.82 |
| BF-98 | Bog Permafrost | 0.5 | −10.97 | −17.87 | −10.77 | −7.03 | −28.32 | 0.30 | 0.73 |
| BF-101 | Bog Permafrost | 0.4 | −10.95 | −18.66 | −10.72 | −7.16 | −20.06 | 0.34 | 0.76 |
| BF-104 | Bog Permafrost | 0.4 | −9.92 | −19.20 | −10.43 | −6.65 | −25.93 | 0.41 | 0.80 |
| BF-105 | Bog Permafrost | 0.3 | −10.20 | −19.15 | −10.53 | −6.81 | −21.06 | 0.43 | 0.81 |
| BF-108 | Bog Permafrost | 0.5 | −8.30 | −17.62 | −8.08 | −4.71 | −29.73 | 0.58 | 0.87 |

**Table 2.** *Cont.*

| Site ID | Site Type | ALT(m) | HH$_{dB}$ | HV$_{dB}$ | VV$_{dB}$ | Span | $\phi_{HH} - \phi_{VV}$ | pmin | pmax |
|---------|-----------|--------|-----------|-----------|-----------|------|------------------------|------|------|
| BF-109 | Bog Permafrost | 0.5 | −10.74 | −21.02 | −11.18 | −7.54 | −21.10 | 0.38 | 0.86 |
| DBF-99 | Deep Swamp Conifer | >1.0 | −9.68 | −15.87 | −9.62 | −5.70 | −32.64 | 0.02 | 0.72 |
| DBF-106 | Bog Permafrost | 1 | −10.10 | −18.59 | −10.12 | −6.26 | −14.51 | 0.29 | 0.73 |
| DBF-107 | Bog Permafrost | 1.2 | −8.81 | −17.94 | −9.60 | −5.63 | −17.50 | 0.36 | 0.81 |
| VDF-CS-96 | Deep Collapse Scar | >1.8 | −9.00 | −18.83 | −9.53 | −5.73 | −27.53 | 0.50 | 0.82 |
| CS-78 | Collapse Scar | N/A | −11.00 | −21.79 | −11.25 | −7.76 | −20.91 | 0.57 | 0.86 |
| CS-89 | Collapse Scar | N/A | −9.12 | −17.18 | −9.41 | −5.64 | −24.22 | 0.09 | 0.78 |
| CS-92 | Collapse Scar | N/A | −8.07 | −16.05 | −9.10 | −4.83 | −15.47 | 0.30 | 0.74 |
| CS-97 | Collapse Scar | N/A | −9.80 | −18.71 | −9.70 | −6.22 | −28.94 | 0.54 | 0.85 |
| CS-100 | Collapse Scar | N/A | −9.92 | −16.65 | −10.37 | −6.25 | −37.42 | 0.08 | 0.72 |
| FC-102 | Forest-Conifer | N/A | −7.70 | −15.62 | −8.72 | −4.45 | −38.36 | 0.10 | 0.73 |

**Table 3.** Touzi Decomposition and Cloude ICTD Parameters.

| Site ID | $\phi_{s1}$ | $\phi_{s2}$ | $\lambda_1$ | $\lambda_2$ | $\lambda_3$ | $\alpha_{s1}$ | $\alpha_{s2}$ | $\alpha_{s3}$ | Cloude $\alpha$ | H |
|---------|-------------|-------------|-------------|-------------|-------------|---------------|---------------|---------------|-----------------|---|
| BF-79 | −36.00 | 72.00 | 0.77 | 0.13 | 0.10 | 6.35 | 39.54 | 19.08 | 28.21 | 0.64 |
| BF-90 | −75.26 | 83.78 | 0.62 | 0.22 | 0.16 | 10.80 | 72.25 | 78.99 | 38.51 | 0.84 |
| BF-91 | −27.72 | 80.78 | 0.71 | 0.17 | 0.12 | 5.26 | 75.18 | 85.12 | 31.10 | 0.72 |
| BF-93 | −43.18 | 74.60 | 0.76 | 0.16 | 0.09 | 8.7 | 24.78 | 80.82 | 32.48 | 0.65 |
| BF-94 | −71.59 | 79.37 | 0.76 | 0.13 | 0.11 | 7.0 | 53.35 | 61.12 | 32.03 | 0.65 |
| BF-98 | −32.64 | 68.18 | 0.61 | 0.23 | 0.15 | 5.00 | 39.52 | 64.34 | 39.49 | 0.84 |
| BF-101 | −82.50 | 83.34 | 0.68 | 0.17 | 0.13 | 10.5 | 75.1 | 83.34 | 37.67 | 0.8 |
| BF-104 | −74.48 | 69.90 | 0.689 | 0.20 | 0.10 | 12.6 | 47.1 | 76.74 | 33.22 | 0.7 |
| BF-105 | −62.00 | 47.00 | 0.70 | 0.19 | 0.10 | 9.75 | 27.4 | 67.65 | 31.48 | 0.7 |
| BF-108 | −59.72 | 80.78 | 0.77 | 0.13 | 0.10 | 8.00 | 50.98 | 83.08 | 30.17 | 0.63 |
| BF-109 | −47.11 | 79.17 | 0.72 | 0.20 | 0.08 | 5.16 | 71.58 | 84.35 | 31.17 | 0.69 |
| DBF-99 | 25.47 | 18.34 | 0.63 | 0.21 | 0.16 | 7.05 | 46.78 | 38.67 | 40.37 | 0.84 |
| DBF-106 | 32.58 | 30.15 | 0.72 | 0.17 | 0.11 | 3.26 | 26.71 | 65.52 | 29.48 | 0.71 |
| DBF-107 | −63.77 | −22.89 | 0.75 | 0.16 | 0.09 | 10.00 | 33.19 | 41.19 | 28.03 | 0.65 |
| VDF-CS-96 | −49.70 | 86.14 | 0.76 | 0.15 | 0.09 | 7.0 | 55.53 | 80.44 | 30.14 | 0.65 |
| CS-78 | −62.65 | −2.17 | 0.81 | 0.12 | 0.07 | 3.53 | 13.46 | 71.15 | 25.05 | 0.54 |
| CS-89 | −76.32 | 60.41 | 0.66 | 0.20 | 0.14 | 8.48 | 22.95 | 60.90 | 35.78 | 0.79 |
| CS-92 | −39.30 | 8.31 | 0.67 | 0.21 | 0.12 | 10.06 | 22.17 | 56.62 | 34.26 | 0.77 |
| CS-97 | −24.89 | 24.17 | 0.74 | 0.15 | 0.10 | 5.47 | 20.82 | 69.61 | 32.14 | 0.67 |
| CS-100 | −58.36 | 4.94 | 0.58 | 0.26 | 0.16 | 14.35 | 21.44 | 28.34 | 43.91 | 0.87 |
| FC-102 | −64.79 | 63.19 | 0.60 | 0.25 | 0.15 | 20.70 | 58.34 | 41.73 | 42.60 | 0.85 |

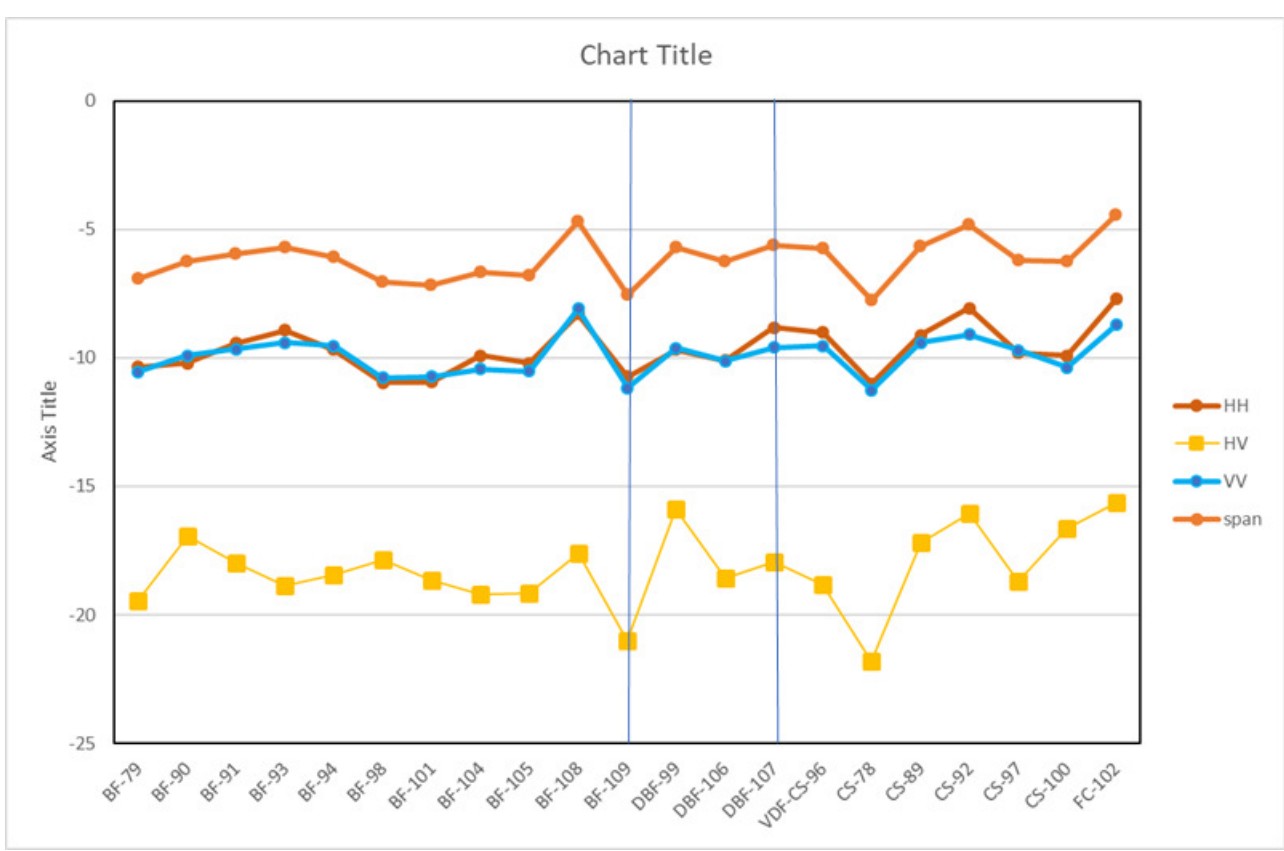

**Figure 34.** Site A: multipolarization channel and span curves (in dB). Sample classes: BF: bog permafrost; DBF: deep bog permafrost; VDF: very deep permafrost; CS: collapse scar; FC: forest conifer.

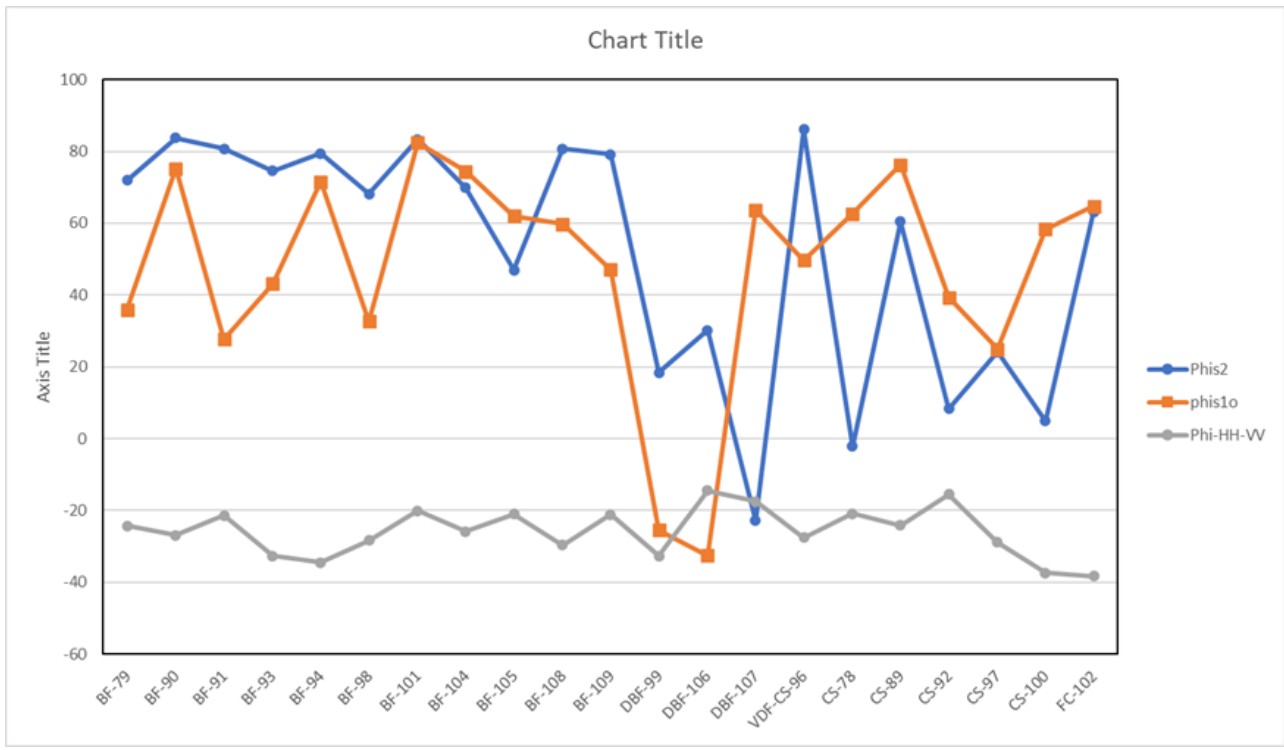

**Figure 35.** Site A: Scattering type phase (Phis1, Phis2) and HH-VV phase difference curves (in degrees). Sample classes: BF: bog permafrost; DBF: deep bog permafrost; VDF: very deep permafrost; CS: collapse scar; FC: forest conifer.

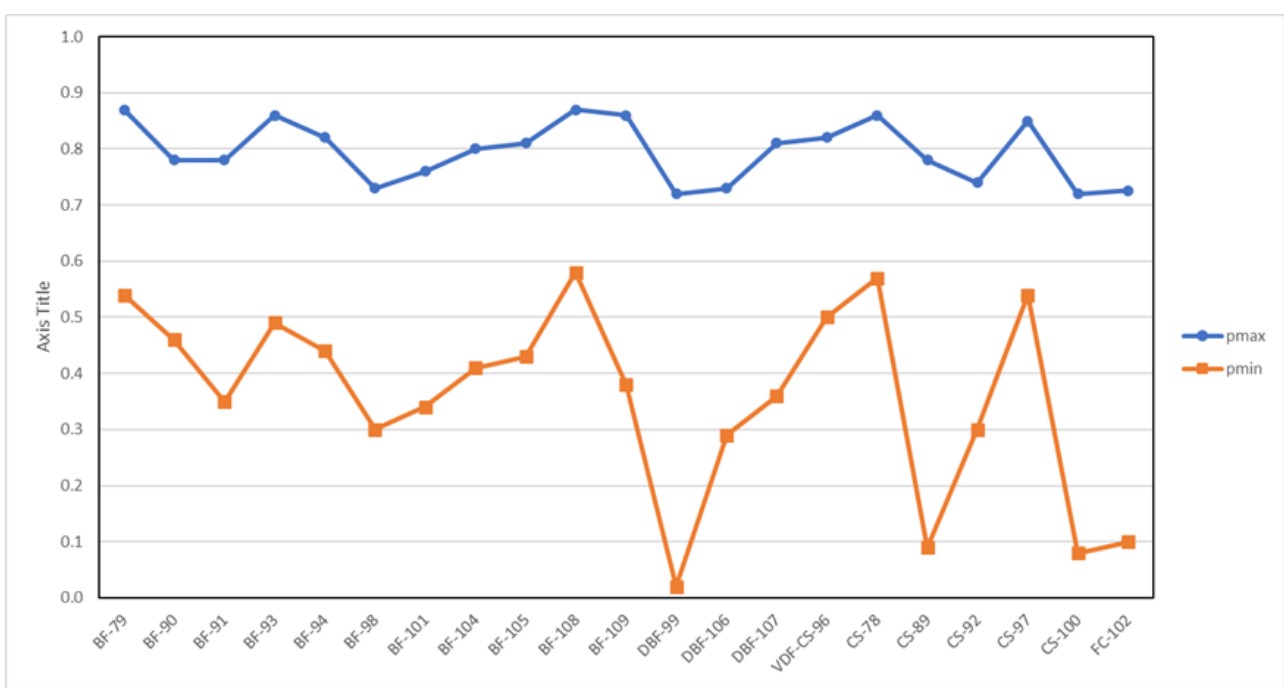

**Figure 36.** Site A: pmax and pmin curves. Sample classes: BF: bog permafrost; DBF: deep bog permafrost; VDF: very deep permafrost (1.8 m and more); CS: collapse scar; FC: forest conifer.

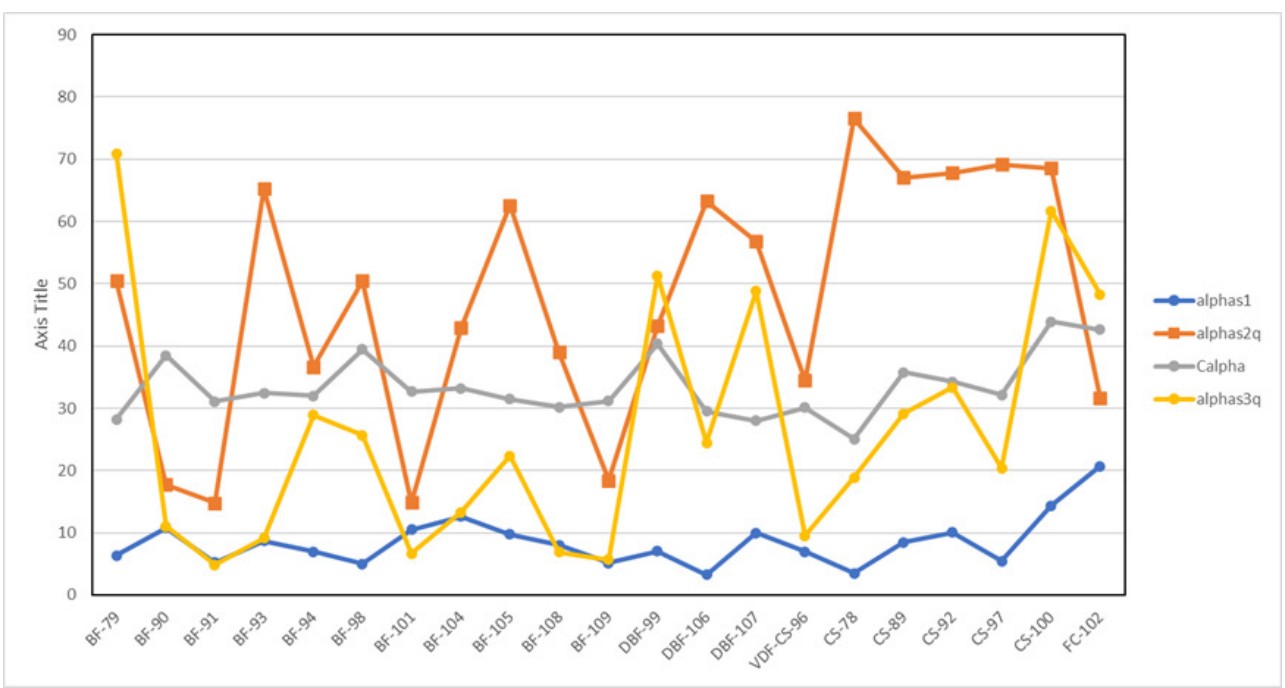

**Figure 37.** Site A: Touzi scattering-type magnitudes and Cloude alpha (in degrees). Sample classes: BF: bog permafrost; DBF: deep bog permafrost; VDF: very deep permafrost; CS: collapse scar; FC: forest conifer.

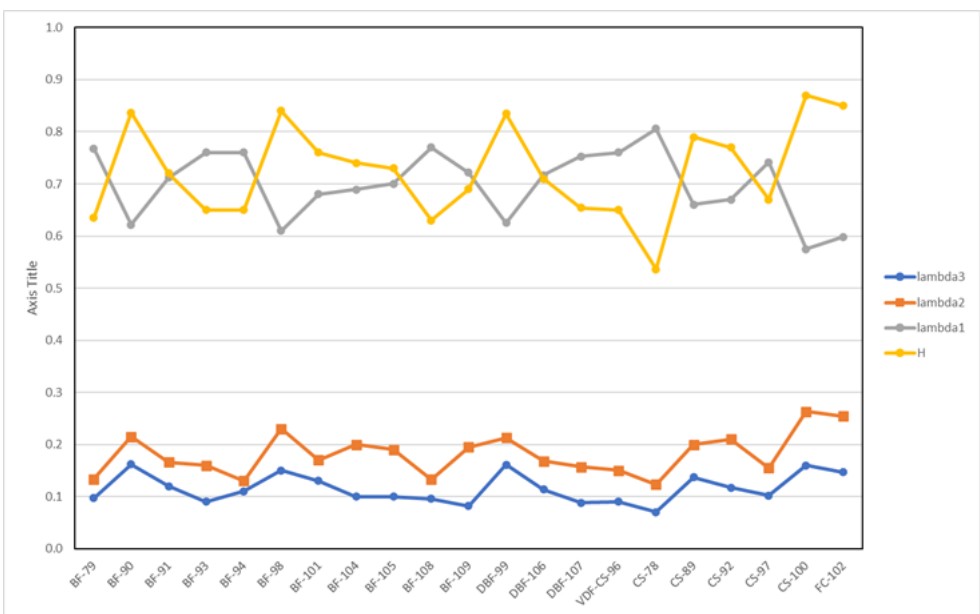

**Figure 38.** Site A: ICTD eigenvalues and entropy. Sample classes: BF: bog permafrost; DBF: deep bog permafrost; VDF: very deep permafrost; CS: collapse scar; FC: forest conifer.

### 5.3.2. PALSAR2 Results: Site B

Figure 39 presents the AGS permafrost classification with the field sample identifications and locations. Figure 40 presents the dominant- and medium-scattering-type phases (opposite) $\phi_{s1o}$ and $\phi_{s2}$. Figures 41 and 42 present the HH-VV phase difference and the minimum DoP (pmin). The curves that represent the multipolarization (HH, HV, VV) intensities and span (in dB) for the various sites are given in Figure 43. The curves of $\phi_{s1o}$, $\phi_{s2}$, and the extrema of the DoP (pmin, pmax) are given in Figures 44 and 45. The medium scattering's Huynen maximum polarization return ($m_2$) is presented in Figure 46. The analysis of the results obtained for site B confirmed the ones obtained at site A, as discussed in the following:

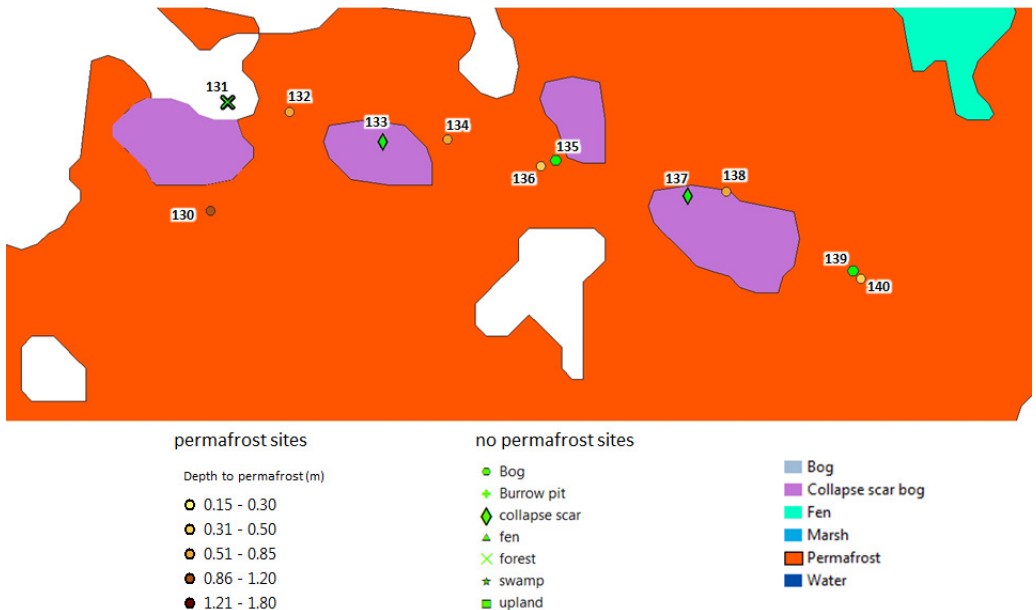

**Figure 39.** Site B: AGS classification.



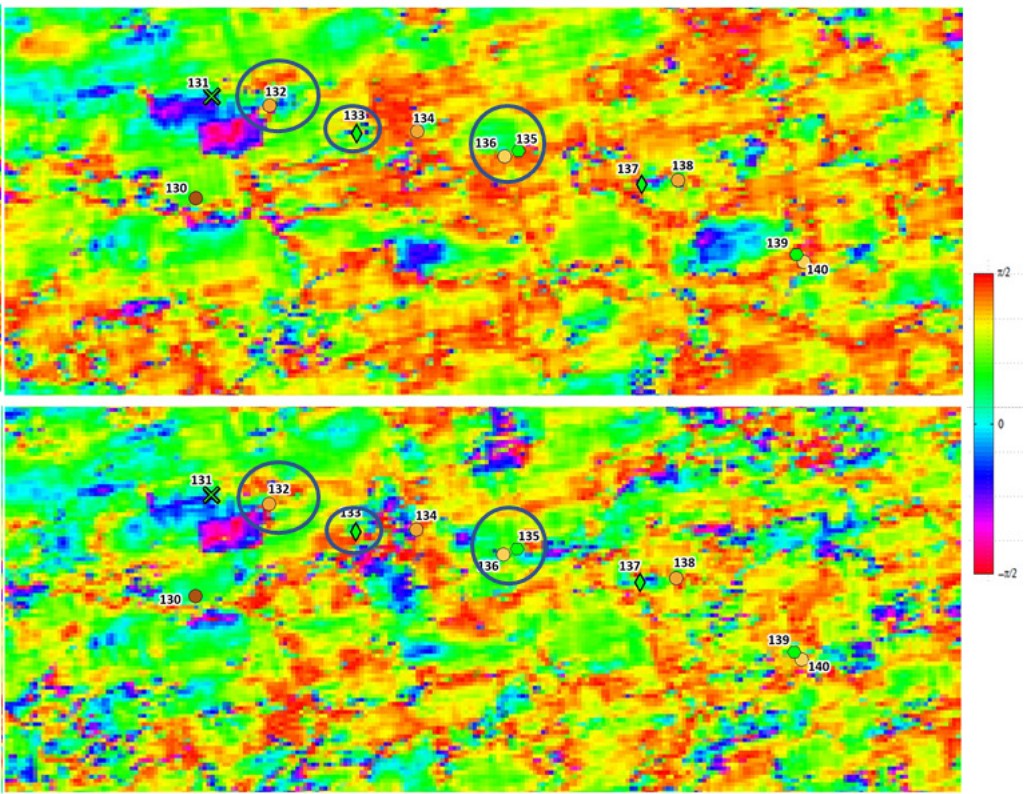

**Figure 40.** Site B: Touzi scattering-type phase: phis1o and Phis2.

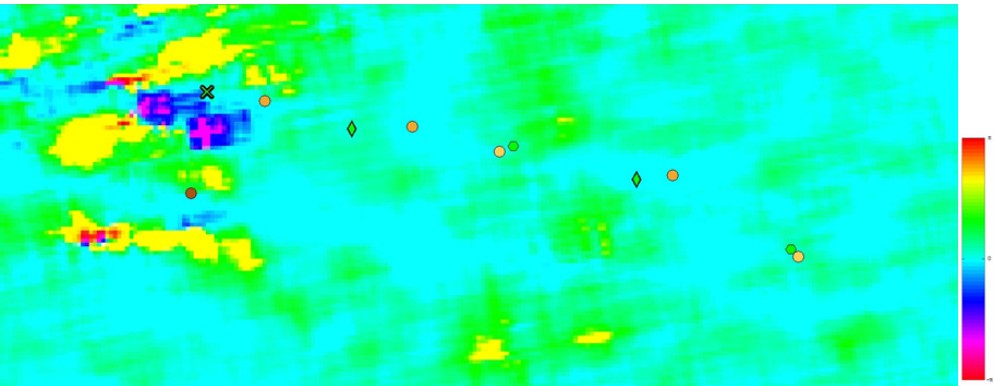

**Figure 41.** Site B: HH-VV phase difference.

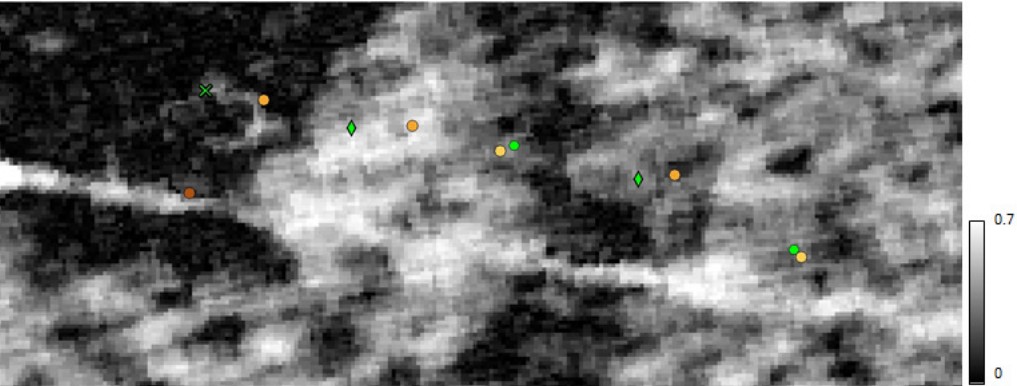

**Figure 42.** Site B: pmin.

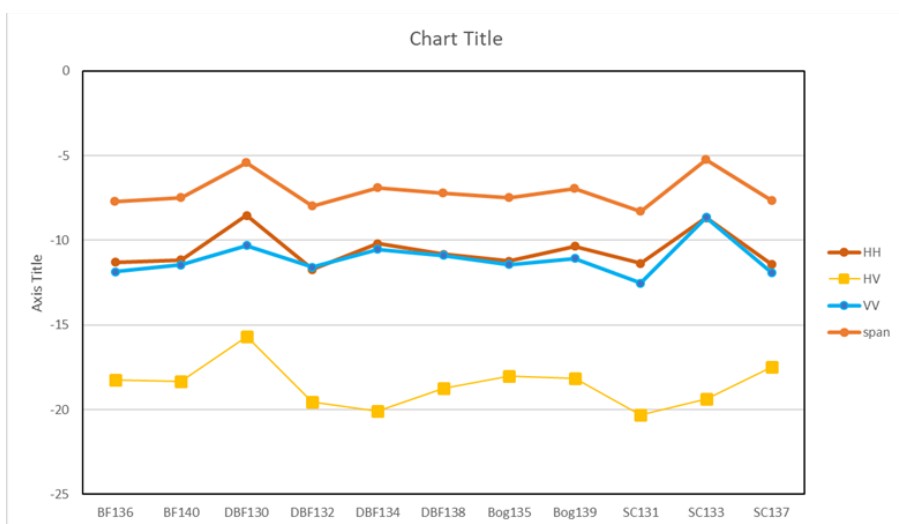

**Figure 43.** Site B: multipolarization and span curves (in dB). Sample classes: BF: bog permafrost; DBF: deep bog permafrost; VDF: very deep permafrost (1.8 m and more); CS: collapse scar; FC: forest conifer.

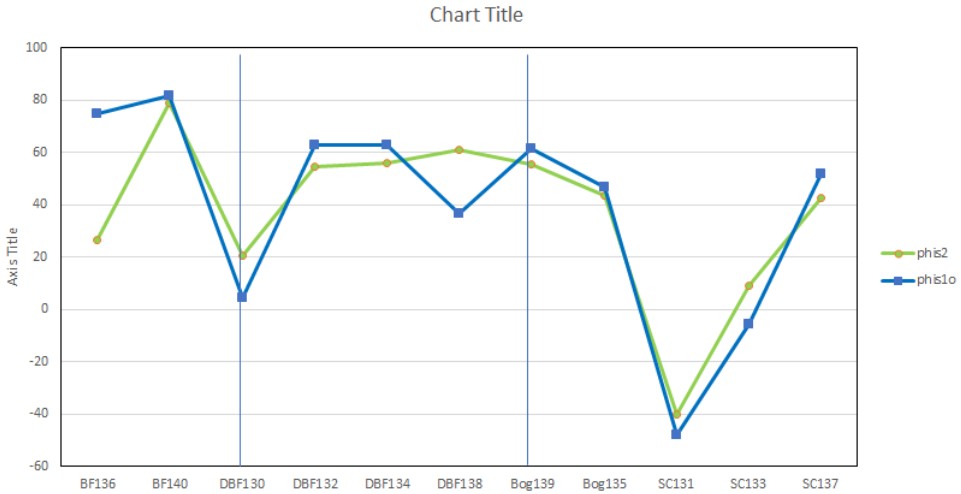

**Figure 44.** Site B: Scattering type phase (Phis1, Phis2) in degrees. Sample classes: BF: bog permafrost; DBF: deep bog permafrost; VDF: very deep permafrost; CS: collapse scar; FC: forest conifer.

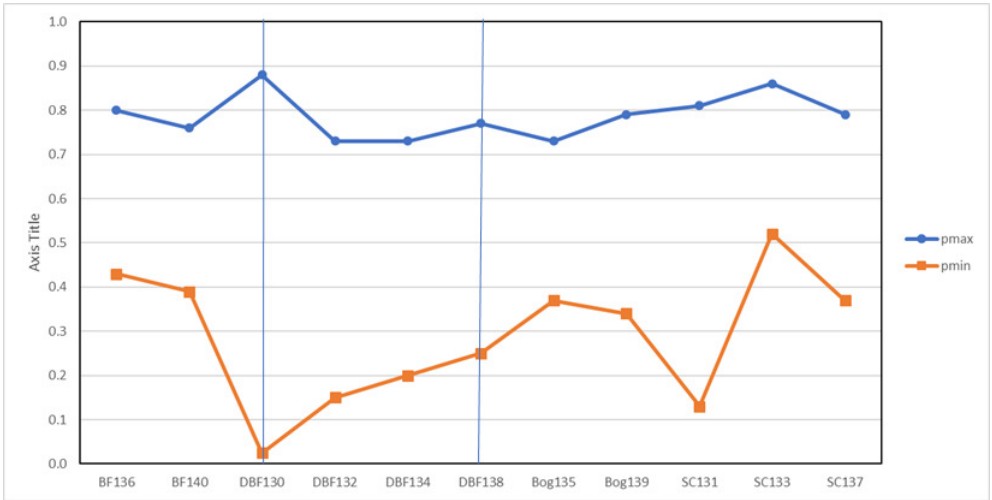

**Figure 45.** Site B: pmin and pmax curves.

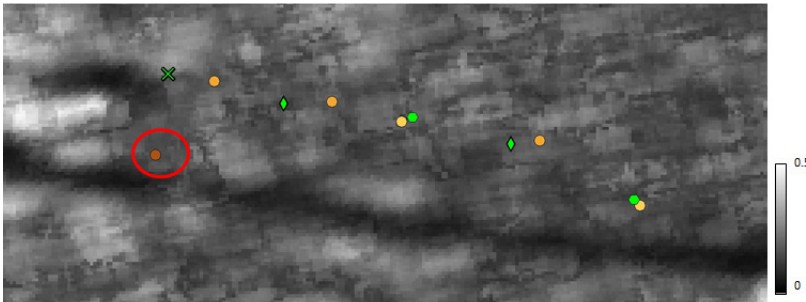

**Figure 46.** Site B: medium scattering's Huynen maximum polarization return ($m_2$).

- $\phi_{s2}$ performed better than $\phi_{s1o}$ for permafrost identification. The results obtained at site BF136 confirmed this important statement, as discussed in the following.
- BF-136: the bog permafrost site was originally assigned to a treed bog underlain by a relatively deep (40 cm) permafrost, according to Figure 39. A detailed analysis of the field data collected by the AGS at that site revealed that permafrost was present in the area but just as small thin patches in otherwise homogeneous looking bog-caribou vegetation. Consequently, that area could not be considered as a treed bog underlain by permafrost. That site was not assigned to the permafrost class according to $\phi_{s2} = 26.63°$, in contrast to $\phi_{s1}$ which misassigned it to the permafrost class with $\phi_{s1} = -74.89°$, according to Figures 40 and 44.
- $\phi_{s2}$ and $\phi_{s1}$ had similar values on the other sites.
- The use of $\phi_{s2}$ permitted the exclusion of all the samples located in areas of very deep permafrost (more than 50 cm) from the permafrost class.
- pmin and $m_2$ could be used (prior to $\phi_{s2}$) to remove eventual scattering-type phase ambiguities and exclude nonpermafrost areas from the permafrost class.

### 5.3.3. Global Analysis of the Study Area

As discussed previously, the study area presented in Figure 18 was dominated by the treed bog class according to the AWI classification of Figure 20 and confirmed by the minimum DoP, pmin, of Figure 25 derived from the recalibrated PALSAR2 image. The medium scattering's maximum polarization return, $m_2$, presented in Figure 46, could be used to remove $\phi_{s2}$ ambiguities that might affect very deep permafrost classification as discussed above.

The comparison of the permafrost area (outlined in Figures 20 and 21) assigned to the permafrost class by AGS (in pink) of Figure 19 and $\phi_{s2}$ (in orange) of Figure 23, showed that the area assigned to relatively deep permafrost was much larger than the permafrost area (of deep and very deep permafrost) assigned by the AGS classification. The area outlined in Figure 19, which was assigned to the bog class by the AWI, was wrongly assigned to the fen class by the LiDAR-Landsat AGS classification. The AGS classification assumption that discontinuous permafrost was located in bog plateaus with extensive caribou lichen [5] might be a limitation that would explain the larger extension of permafrost area detected (up to 50 cm) by $\phi_{s2}$ in Figure 23. The inability of the combination of LiDAR and Landsat to accurately identify treed bogs led to an automatic exclusion of this treed-bog area from the permafrost class, as seen in Figure 19. Unfortunately, no field data were collected in that extended treed bog for the validation of the results obtained with $\phi_{s2}$. This will be conducted in the near future, jointly with the AGS, in conjunction with PALSAR2 image acquisitions that will be collected at the study site during the field data collection.

It is worth noting that the polarimetric results obtained over the treed-bog area, outlined in Figures 20 and 21, were significantly improved by the ALOS2 recalibration described in Section 4, as can be seen in Figures 12–15. The treed-bog area cross-polarizations (HV and VH), which were much lower (about 10 dB) than the copolarizations HH and VV, were largely contaminated (in the original image) by the miscellaneous ALOS2 image calibration. The image recalibration led to pure HV and VV (in addition to HH and

VV) measurements, and this permitted the full exploitation of the excellent ALOS2 NESZ ($-37$) for an enhanced sensitivity of long-penetrating polarimetric ALOS2 wave data to subsurface permafrost underlying peatlands regions.

## 6. Conclusions

The validation with field data of the results obtained in the Namur Lake study site revealed that the dominant- and medium-scattering-type phases, $\phi_{s1o}$ and $\phi_{s2}$, derived using the Touzi decomposition [16,23] were the most sensitive polarimetric parameters to peatland subsurface discontinuous permafrost (up to 50 cm). The multipolarization (HH, HV, VV) intensity, the HH-VV phase difference, the Cloude–Pottier $\alpha$-H and all the other parameters generated by the Touzi decomposition could not identify permafrost samples and separate them from the ones located in nonpermafrost areas. $\phi_{s2}$ performed better than $\phi_{s1o}$ and led to an accurate identification of permafrost samples in peatland areas underlain by relatively deep permafrost (up to 50 cm). The use of the additional information provided by the medium scattering's maximum polarization return $m_2$ and the minimum DoP (pmin) permitted the solving for $\phi_{s2}$ ambiguities that may occur in areas with very deep permafrost (deeper than 50 cm).

These very promising results were obtained with polarimetric ALOS2 thanks to its excellent performance in term of NESZ ($-37$ dB). We previously showed in [21] that the excellent results obtained with ALOS for peatland classification would not have been obtained if ALOS's NESZ had not been better than $-34$ dB. The excellent ALOS2 NESZ ($-37$ dB) permits the deep penetration (up to 50 cm) of the ALOS2 wave under the peat surface for the enhanced detection and characterization of discontinuous permafrost in peatland regions. The recalibration of ALOS2's beam FP6-4 permitted a significant enhancement of permafrost detection and a full exploitation of the excellent ALOS2 NESZ capabilities for the enhanced identification of permafrost areas and their accurate separation from areas that were not underlain by permafrost. It is worth noting that the miscellaneous problem with the ALOS2 FP6-4 mode has recently been solved by JAXA, and an updated list of the polarimetric calibration parameters (transmitter–receiver distortion matrix, as well as channel imbalances) that has been used since January 2018 has been provided [52].

The polarimetric ALOS2 detection of relatively deep (up to 50 cm) subsurface permafrost was recently confirmed in peatland regions along the Inuvik-Tuktoyuktuk highway (Northwest Territories, Canada) using (recalibrated) polarimetric ALOS2 images collected in August 2017 [60,61]. Very long penetrating P-band polarimetric SAR images were collected in the same week by the NASA airborne AIRMOSS, in the context of the Arctic-Boreal Vulnerability Experiment (ABoVE) organized by NASA [62]. The validation of the results obtained with polarimetric L-band ALOS2 and P-band AIRMOSS data, using the field data collected during the campaign, confirmed the relatively deep penetration of ALOS2 (up to 50 cm) and the very deep penetration (up to 1 m) of P-band AIRMOSS [60].

In the near future, new ALOS2 campaigns will be organized, jointly with JAXA, on the Namur Lake site for further validation of the very promising ALOS2 polarimetric information in support of an enhanced mapping of discontinuous permafrost. Ground-penetrating radar [63] will be collected, in addition to the field sampling conducted in 2014, for a better identification of subsurface discontinuous permafrost distribution. Further field data collection will be conducted in 2024 and 2025 to assess the potential of the upcoming ALOS4 for operational discontinuous permafrost mapping and monitoring. ALOS4, which is equipped with digital antenna beaming [64,65] and which is planned to have an excellent NESZ (like ALOS and ALOS2), should permit a large-cover-high-resolution imaging of Northern Alberta for the operational use of polarimetric ALOS4 in support of an enhanced mapping and monitoring of the discontinuous permafrost distribution in Northern Alberta and in other regions of Northern Canada.

**Author Contributions:** Conceptualization, R.T.; methodology, R.T. and M.S.; software, R.T.; validation, R.T., P.W., X.J., M.H. and S.M.P.; formal analysis, P.W., X.J. and M.H.; investigation, R.T., P.W., X.J., M.H., S.M.P. and M.S.; resources, CCRS; data curation, JAXA is thanked for having provided ALOS2 data; writing—original draft preparation, R.T., P.W., M.S. and S.M.P.; writing—review and editing, R.T. and P.W.; visualization, P.W.; supervision, R.T.; project administration, R.T.; funding acquisition, Canadian Space Agency. All authors have read and agreed to the published version of the manuscript.

**Funding:** This research received no external funding.

**Data Availability Statement:** The ALOS-2 original data are provided by JAXA under the Research Announcement on the Earth Observations (EO-RA).

**Conflicts of Interest:** The authors declare no conflict of interest.

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
