# Peer review of "Polarimetric L-Band ALOS2-PALSAR2 for Discontinuous Permafrost Mapping in Peatland Regions"

_remotesensing, doi:10.3390/rs15092312_

Round 1

Reviewer 1 Report

The manuscript is sufficient to me as a new one and logically and scientifically based. I am not agreed with sentence "LiDAR is not a suitable methodology for repeated mapping and/or monitoring ongoing permafrost changes at large scale", because it is quite good for air mapping (http://www.riegl.com/nc/products/airborne-scanning/produktdetail/product/scanner/70/ ) of large scale objects such as Alps glaciers et ctr. Meanwhile it is also be good for validation of proposed technology.

Author Response

Dear Reviewer,

Thank for having reviewed our paper and for the helpful comments.

I totally agree with you on the comment you have made on the Lidar. I do confess that our mention on LIDAR  you have referred too is very confusing. For sure, it not very convenient and it is very expansive to have a yearly LIDAR campaign for permafrost. But still  right now the LIDAR is the only operationel tool that provides the required  information for  discontinuous permafrost mapping. 

To avid any confusion, the following sentence you have referred to (2nd paragraph of the Main Introduction, "However, LiDAR is not a suitable methodology for repeated mapping and/or monitoring ongoing permafrost changes at large scale, and cannot assess important indicators of
permafrost thermal state such as active layer thickness (ALT)."

has been erased from the Main Introduction.

Thank you again for the helpful comments and having appreciated our work.

Ridha Touzi

Reviewer 2 Report

This article is an extension of the conference paper presented at IGARSS 2019. It delves deeper into the application of the Touzi decomposition technique on the re-calibrated polarimetric ALOS2 data collected in peatland regions. Additionally, it provides more detailed information on the AGC field data.

Generally, the data analysis is thorough and the results are clearly presented. My only suggestion is that all images and maps must have a scale, a north arrow, and coordinates that mark the geo-location.

Author Response

Dear Reviewer,

Thank you for having reviewed our paper and your very helpful comments.

"all images and maps must have a scale, a north arrow, and coordinates that mark the geo-location".

WE have adopted your comment on the revised version.  The image scale, the
four cardinal directions (N, S, E, W) directions, and the coordinates that mark the geo-location are added in all images used as input to the analysis; ALOS2 multi-polarization image, AGS permafrost classification, AGCC and AWI Classification. We did that for Figure1, 2, 3 and 4 on the images that provide a Global view of the area investigation, and for Figures 18, 19, 20 and 21 that present the study site.

Thank you again for the very helpful comment

Best regards

Ridha

Reviewer 3 Report

The Authors' Polarization Measurements of Discontinuous Permafrost Mapping in the Peatland Region Using L-Band ALOS2-PALSAR2. There is no doubt that this work is rewarding, however, problems with the organization and writing of the manuscript resulted in a lack of clarity about what was to be presented in the manuscript, so I think the author needs to make one major revision, then I will give more specific comments based on the revised manuscript. The main comments that need to be revised at this time are:

(1)   I strongly recommend that the author add a flow chart to summarize the work of the entire manuscript so that the reader can clearly understand the meaning of the manuscript.

(2)   Abstract section: The abstract section does not summarize well, it is more like the result, such as “It is shown that PALSAR2 image, ...”.

(3)   Abstract section: The abstract should primarily describe the content of the manuscript and should not normally contain literature citations.

(4)   Key words: I didn't find the key words in the manuscript, did I miss them?

(5)   Introduction: “average annual temperature”, What temperature is it? Air/surface/ground surface/ground?

(6)   Research Area: The author should carefully draw a map of the distribution of the study area and its location, because for most readers, they don’ t know where the study area is.

(7)   Picture: There are many problems with the pictures, such as blurred legends, no units and some pictures even missing legends.

Author Response

Dear Reviewer,

Thank you very much for having reviewed our paper and for the very helpful comments and suggestions.

Abstract and Main Introduction; 

  1. The Abstract of new paper version is more detailed and provides a more clear picture of the work completed in this study.
  2. The last paragraph of the Main Introduction includes a more detailed presentation of  all the Sections, the subsections, and the issues that are going to be discussed.
  3. With this very detailed Introduction, we would not think that it is necessary to include a FlowChart that would present the paper. During my 30 years of carrier in SAR, as a Research Scientist, I has never seen a journal paper that uses a Flowchart to present the Content of the paper. 
  4.   

    WE have adopted your comment on the revised version.  The image scale, the four cardinal directions (N, S, E, W) directions, and the coordinates that mark the geo-location are added in all images used as input to the analysis; ALOS2 multi-polarization image, AGS permafrost classification, AGCC and AWI Classification. We did that for Figure1, 2, 3 and 4 on the images that provide a Global view of the area investigation, and for Figures 18, 19, 20 and 21 that present the study site.

  5. Pictures; the missing legends and units have been added in the revised version. Higher definition pictures are provided to the journal to avoid any eventual blurring.

Many thanks again for your very helpful comments and suggestions.

Best regards

Ridha